# Study protocol: Minimum effective low dose: anti-human thymocyte globulin (MELD-ATG): phase II, dose ranging, efficacy study of antithymocyte globulin (ATG) within 6 weeks of diagnosis of type 1 diabetes

Charlotte S Wilhelm-Benartzi [ID],[1] Sarah E Miller,[2,3] Sylvaine Bruggraber,[2] Diane Picton,[2] Mark Wilson,[2] Katrina Gatley,[3] Anita Chhabra,[4] M Loredana Marcovecchio,[2] A Emile J Hendriks,[2] Hilde Morobé,[5] Piotr Jaroslaw Chmura,[6] Simon Bond,[3] Bärbel Aschemeier-Fuchs,[7] Mikael Knip,[8,9] Timothy Tree,[10] Lut Overbergh,[5] Jaivier Pall,[11] Olivier Arnaud,[12] Michael J Haller,[13] Almut Nitsche,[14] Anke M Schulte,[14] Chantal Mathieu,[5] Adrian Mander,[1] David Dunger [ID] [2,15]

For numbered affiliations see end of article.

**Correspondence to**
Dr Charlotte S Wilhelm-Benartzi;
Wilhelm-BenartziC@cardiff.ac.uk

## ABSTRACT

**Introduction** Type 1 diabetes (T1D) is a chronic autoimmune disease, characterised by progressive destruction of the insulin-producing β cells of the pancreas. One immunosuppressive agent that has recently shown promise in the treatment of new-onset T1D subjects aged 12–45 years is antithymocyte globulin (ATG), Thymoglobuline, encouraging further exploration in lower age groups.

**Methods and analysis** Minimal effective low dose (MELD)-ATG is a phase 2, multicentre, randomised, double-blind, placebo-controlled, multiarm parallel-group trial in participants 5–25 years diagnosed with T1D within 3–9 weeks of planned treatment day 1. A total of 114 participants will be recruited sequentially into seven different cohorts with the first cohort of 30 participants being randomised to placebo, 2.5 mg/kg, 1.5 mg/kg, 0.5 mg/kg and 0.1 mg/kg ATG total dose in a 1:1:1:1:1 allocation ratio. The next six cohorts of 12–15 participants will be randomised to placebo, 2.5 mg/kg, and one or two selected middle ATG total doses in a 1:1:1:1 or 1:1:1 allocation ratio, as dependent on the number of middle doses, given intravenously over two consecutive days. The primary objective will be to determine the changes in stimulated C-peptide response over the first 2 hours of a mixed meal tolerance test at 12 months for 2.5 mg/kg ATG arm vs the placebo. Conditional on finding a significant difference at 2.5 mg/kg, a minimally effective dose will be sought. Secondary objectives include the determination of the effects of a particular ATG treatment dose on (1) stimulated C-peptide, (2) glycated haemoglobin, (3) daily insulin dose, (4) time in range by intermittent continuous glucose monitoring measures, (5) fasting and stimulated dry blood spot (DBS) C-peptide measurements.

## Strengths and limitations of this study

► Prior evidence of efficacy and safety in this condition.
► Alignment to a standard evaluation of efficacy and mechanistic outcomes.
► Large clinical infrastructure to support recruitment and study integrity across the European Union and the UK.
► Limited long-term safety and efficacy data.

**Ethics and dissemination** MELD-ATG received first regulatory and ethical approvals in Belgium in September 2020 and from the German and UK regulators as of February 2021. The publication policy is set in the INNODIA (An innovative approach towards understanding and arresting Type 1 diabetes consortium) grant agreement ( www.innodia.eu).
**Trial registration number** NCT03936634; Pre-results.

## INTRODUCTION
### Background

Type 1 diabetes (T1D) is a chronic autoimmune disease, characterised by progressive destruction and dysfunction of the insulin-producing β cells of the pancreas. The incidence of T1D is increasing and around 130 000 children and adolescents worldwide will develop T1D each year.[1–3] A diagnosis of T1D is associated with a lifelong dependence on insulin therapy. The primary goal of therapy is to maintain glycaemic control as assessed by glycated haemoglobin (HbA1c), in

a range sufficient to prevent long-term renal, cardiac and retinal complications as shown in the Diabetes Control and Complications Trial.[4] However, recent data arising from continuous glucose monitoring (CGM) suggests that as well as avoidance of hypoglycaemia, improved 'time in target' may also be an important patient-related outcome.[5 6] Technological advances and the increasing uses of insulin pumps and glucose sensors are gradually having an impact on glycaemic trends, but few people achieve glycaemic targets of <7%–7.5% (<53–58 mmol/mol) recommended by recent guidelines to prevent long term complications.[7]

Development of T1D is triggered by pancreatic islet-specific CD4 + and CD8+T cells, as well as B cells, which target and eliminate insulin-producing β cells. Recent epidemiological data indicate that the pathogenic mechanisms which lead to destruction of β cells may already begin soon after birth and by the time of diagnosis (peak between 5 and 15 years) residual functional β cell mass may be reduced already by around 50%.[8 9] Although still a subject of investigation, there is no evidence that even intensive insulin therapy starting right after T1D diagnosis will be able to prevent further deterioration in β cell function. Paediatric patients will lose up to 50% of residual β cell function over the first year from diagnosis, while adults lose about 15%.[10] Thereafter, decline in β cell function is highly variable and evidence of persisting active β cell function may be observed many years later particularly in those diagnosed during adult life.[11–13] Preservation of some β cell function, as assessed by a Mixed Meal Tolerance Test (MMTT), is associated with improved glycaemic control, reduction in hypoglycaemia risk and significant reductions in the risk for progression of retinopathy and other long-term complications.[14 15]

The first attempts to preserve β cell function through immunosuppression treatment included the use of ciclosporin in the 1980s.[16 17] In recent years, a wide range of agents which affect the regulation of the immune system have been explored, either addressing genetically validated pathways associated with T1D risk[18 19] or those repositioned following proven efficacy in other autoimmune conditions.[20 21] One immunosuppressive agent that has recently shown promise in preservation of β cell function in new-onset T1D is rabbit anti-human thymocyte immunoglobulin (antithymocyte globulin (ATG); Thymoglobuline), a polyclonal antithymocyte human thymocyte agent, indicated for various indications in the US and Europe. It's main indication is the prevention and treatment of organ transplantation rejection when used in a dose range of 6.5–21 mg/kg ATG.[22] Recent clinical data have now demonstrated that ATG may be effective at a much lower dose of 2.5 mg/kg in preventing β cell functional decline in newly diagnosed T1D.[23–26]

## Pharmacokinetics and pharmacodynamics of ATG
Pharmacokinetics of ATG have been studied in patients undergoing organ and stem cell transplantations.[22] ATG is a selective immunosuppressive agent, mostly acting on T lymphocytes and lymphocyte depletion probably constitutes the primary mechanism of the immunosuppression induced by ATG. This depletion is both peripheral and central. Peripheral lymphocyte depletion can be detected as early as 24 hours after the first infusion whereas lymphocyte counts start to rise as soon as ATG is discontinued. This lymphocyte depletion has been shown to occur in vitro by several different mechanisms (eg, apoptosis, complement dependent lysis and antibody dependent cytotoxicity) but the exact mechanisms which take place in vivo remain undefined. In addition to T-cell depletion, ATG has effects on dendritic cells, causing their apoptosis in vitro and ATG does not activate B cells. ATG targets a number of different cell surface epitopes (eg, CD3, CD7, CD8, CD19, CD20, CD32, CD28), causing their down regulation. The over 40 epitopes targeted by the polyclonal rabbit IgG ATG include those involved in immune response, apoptosis and signal transduction, and includes both B-cell and T-cell epitopes. In the context of T1D, the ability of ATG to target multiple T-cell antigens, including targets used in previous T1D trials with monotargeted drugs, represents a unique multi targeted therapy opportunity that may has the potential to promote tolerogenic responses and blocks autoimmunity with one single multitargeted drug dose.[27]

## Data from preclinical T1D studies
Preclinical data have reported potential efficacy of murine ATG (mATG) in the non-obese diabetic T1D mouse model in a way that mATG treatment attenuated the development of autoimmune diabetes in an age-dependent manner.[28] In this model, the treatment was only efficacious at onset of the disease or in the late prediabetic phase (12 weeks of age).[28] Immune analyses suggested that an induction of immunoregulation, rather than simple lymphocyte depletion, contributed to the therapeutic efficacy of mATG and, that there was an early time-dependent window for the ability to delay or reverse autoimmune diabetes, through the enhancement of the functional activity of regulatory T cells.

## Clinical data in T1D
The efficacy of low dose ATG (Thymoglobuline) in T1D has been previously reported in three clinical studies where ATG was used as monotherapy as well as in combination therapy with Granulocyte colony-stimulating factor (G-CSF). In one clinical study, a single low dose ATG (2.5 mg/kg) was given to participants with established T1D (diagnosis 4–24 months; age range 12–45 years) in combination with G-CSF (36 mg) which resulted in C-peptide preservation for up to 18 months, and improved HbA1c[23 24] and the 5-year follow-up data just published.[29] Furthermore, in another clinical trial in participants with new-onset T1D (within 100 days from diagnosis; age range: 12–45 years), a single low dose ATG (2.5 mg/kg) alone vs its combination with G-CSF (36 mg in adults and adjusted for children <45 kg who then received 100 mg/kg per dose) resulted in an 18 months

lasting beneficially therapeutic outcome, as defined by C-peptide preservation and HbA1c improvement.[26] Most recently, Haller *et al* reported a 1-year and 2-year clinical trial outcome data confirming a sustained effect of low dose ATG (2.5 mg/kg) treatment on C-peptide preservation and HbA1c improvement.[25] In contrast, in the trial the Study of Thymoglobulin to ARrest T1D, a higher dose of ATG (6.5 mg/kg) treatment in participants with new-onset T1D failed to achieve β-cell functional preservation.[30 31] The dose dependent differences in β-cell preservation capacity of ATG encourage further investigation to determine the lowest effective dose of ATG in the treatment of new-onset T1D.

ATG safety data in paediatric population are limited but indicate no fundamentally difference to that seen in adults[22] as the treatment is applied per body weight. Thymoglobulin adverse events (AEs) are generally manageable or reversible. AEs previously reported with low dose (2.5 mg/kg) ATG treatment in new-onset T1D (age 12–45 years) participants include immune system disorders such as cytokine release syndrome (CRS-with symptoms of fever, headaches, nausea), lymphopaenia and serum sickness. The latter is the consequence of the rabbit IgG nature of ATG. Most reported symptoms may occur seven to fifteen days after onset of treatment. Slow intravenous drug infusion and premedication (eg, hydrocortisone, heparin, corticosteroid, antihistamine, paracetamol) have been shown to manage those symptoms.[22 26]

The minimal effective low dose (MELD)-ATG trial is run on top of the INNODIA Master Protocol within the INNODIA clinical trial network (www.INNODIA.eu). The MELD-ATG trial intends to assess the safety, tolerability and efficacy of 2.5 mg/kg ATG and in even lower doses in paediatric, adolescent and adult participants with new-onset T1D. In addition, multiple comprehensive molecular assessments are scheduled to further elucidate the mechanisms of action of ATG in paediatric, adolescent and adult patients with new-onset T1D.

## METHODS AND ANALYSIS
### Objectives
The primary objective of the MELD-ATG (rabbit) (MELD-ATG) trial is to first determine the changes in stimulated C-peptide response over the first 2 hours of an MMTT at 12 months for the 2.5 mg/kg ATG arm versus placebo in paediatric, adolescent and adult people with new-onset T1D.

Conditional on finding a statistical difference between the 2.5 mg/kg ATG arm and placebo, the primary objective is also to identify the minimally effective dose (lowest dose significantly different from placebo) among the doses studied in the trial using change in stimulated C-peptide response over the first 2 hours of an MMTT at 12 months versus placebo.

The secondary objectives are to determine the effects of ATG treatment on (1) stimulated C-peptide (baseline, 3, 6, 12 months), (2) HbA1c (baseline, 3, 6, 12 months),

insulin requirements throughout the study, (3) time in range by intermittent CGM (3, 6 and 12 months) measures, (4) fasting and stimulated DBS C-peptide measurements (monthly), (5) mediating a reduction in CD4-positive T cells by relative preservation of CD8-positive T cells and (6) a descriptive analysis of the safety and pharmacodynamic activity profile of different low doses of ATG in different age groups. The exploratory objectives are to study the effects of low doses of ATG treatment on biomarkers related to immunological changes and β-cell death/survival in this population.

Table 1 reports the specific trial objectives and related outcome measures.

### Study summary
MELD-ATG is a phase 2, multicentre, randomised, double-blind, placebo-controlled, multiarm parallel cohort trial. The MELD-ATG trial design is shown in figure 1.

Participants newly diagnosed with T1D within 3–9 weeks of planned treatment day 1 will be recruited sequentially into seven different cohorts. MELD-ATG is an international multicentre trial. The trial will include suitable sites across Europe who are part of the existing INNODIA T1D clinical consortium (www.innodia.eu) and that are confirmed suitable for undertaking this specific trial through an accreditation process. Currently, 12 clinical sites across 10 European countries are planned to be incorporated, and include Belgium, the UK, Austria, Denmark, Finland, Germany, Poland, Slovenia and Italy. Further details on participating sites can be obtained from the MELD-ATG Coordinating team contact (MELD-ATG@medschl.cam.ac.uk) and via the INNODIA web page (Innodia.eu - Clinical Trials).

### Intervention summary
The total dose of ≤2.5 mg/kg ATG will be split into two separate part-doses for intravenous administration over two consecutive days. On treatment day 1, the first infusion of 0.5 mg/kg or 0.1 mg/kg ATG, or placebo of the same volume, will be given over a minimum of 12 hours. On treatment day 2, the second part-dose will be infused over a minimum of 8 hours, commencing between 12 and 24 hours after the first part-dose. A maximum of 60 hours from the start of the first infusion will be allowed to complete all trial drug administration, otherwise the infusion will be discontinued, and no further trial drug will be administered. The trial pharmacist will be responsible for calculating the dose for the participants according to their weight and to the randomisation cohort the participant has been allocated to via the electronic randomisation system, as detailed in the below section 'Randomisation and blinding'. The weight of the participant at the baseline visit will be used to calculate the dose amount which can be rounded up or down to the nearest ml. The placebo (0.9% sodium chloride solution) will be sourced from local hospital stock. There will be an identical volume of infusion solution between ATG

**Table 1** Study objectives and outcomes

| Objectives | Outcome measures | Time point(s) of evaluation of this outcome measure |
|---|---|---|
| **Primary objectives** | | |
| 1. To determine the changes in stimulated C-peptide response over the first 2 hours of an MMTT for 2.5 mg/kg ATG arm vs the placebo. | 1. The area under the stimulated C-peptide response curve over the first 2 hours of an MMTT for the 2.5 mg/kg ATG and placebo arms | 1. 12 months |
| 2. Conditional on finding a statistical difference between the 2.5 mg/kg ATG arm and placebo, to identify the minimally effective dose significantly different to placebo among the doses studied in the trial using change in stimulated C-peptide response over the first 2 hours of an MMTT vs placebo | 2. The area under the stimulated C-peptide response curve over the first 2 hours of an MMTT for the 0.1 mg/kg ATG, 0.5 mg/kg ATG, 1.5 mg/kg ATG, 2.5 mg/kg ATG and placebo arms | 2. 12 months |
| **Secondary objectives** | | |
| 1. To determine the effects of ATG treatment on stimulated C-peptide | 1. The area under the stimulated C-peptide response curve over the first 2 hours of an MMTT measured throughout the study in different ATG dosage groups | 1. Baseline, 3, 6, 12 months |
| 2. To determine the effects of ATG treatment on HbA1c | 2. HbA1c measurements measured throughout the study in different ATG dosage groups in mmol/mol | 2. Baseline, 3, 6, 12 months |
| 3. To determine the effects of ATG treatment on intermittent CGM measures | 3. CGM measures (time in range, time above, time below) measured throughout the study in all different ATG dosage groups and placebo | 3. For 14 days at 3, 6, 12 months |
| 4. To determine the effects of ATG treatment on fasting and stimulated DBS C-peptide measurements | 4. DBS C-peptide measurements measured throughout the study in all different ATG dosage groups and placebo in pmol/L at 0 min and 60 min | 4. Baseline, monthly |
| 5. To determine whether ATG treatment mediates a reduction in CD4-positive T cells but relative preservation of CD8-positive T cells | 5. CD4-positive T cells and CD8-positive T cells measured throughout the study in different dosage groups | 5. Baseline, 1, 2, 4 weeks and 3, 6, 12 months |
| 6. Descriptive analysis of the safety profile of different doses of ATG in different age groups | 6. Safety will be assessed at each visit by physical examination, including assessment of the most commonly reported reactions to ATG, namely serum sickness, CD4 + lymphocyte decrease, cytokine release syndrome, fever, influenza-like symptoms and rash; vital signs (temperature, blood pressure, heart rate); weight, abnormal laboratory parameters (liver, kidney function, full blood count); reporting of adverse events in different dosage groups | 6. Baseline, treatment day 1 and 2, 1, 2, 4 weeks and 3, 6, 12 months |
| 7. To determine the effects of ATG treatment on T1D-associated autoantibodies (GADA, IAA, IA-2A and ZnT8A) | 7. T1D-associated autoantibodies (GADA, IAA, IA-2A and ZnT8A) at the start and end of the study | 7. Screening, 12 months |
| 8. To determine the effects of ATG treatment on insulin requirements | 8. Insulin requirements measured throughout the study in all different ATG dosage groups | 8. Baseline, 1, 2, 4 weeks and 3, 6, 12 months |
| **Exploratory objectives** | | |

Continued

**Table 1** Continued

| Objectives | Outcome measures | Time point(s) of evaluation of this outcome measure |
|---|---|---|
| 1 and 2 To study the effects of treatment on other biomarkers related to immunological changes and β-cell death or survival in this population | 1. The effects of ATG treatment on other biomarkers related to immunological changes and β-cell death or survival in this population | 1. Baseline, 1, 2, 4 weeks and 3, 6, 12 months |
| | 2. Multidimensional analyses of changes in T1D phenotypes by immunological, transcript and mi/small RNA profiling, proteomic, metabolomic and lipidomic studies and the relation of these to clinical outcomes and progression, with the intention of facilitating biomarker discovery, surrogate marker development and potentially participant stratification in future trials | 2. Baseline, 1, 2, 4 weeks and 3, 6, 12 months |

ATG, antithymocyte globulin; CGM, continuous glucose monitoring; DBS, dried blood spot; GADA, Glutamic acid decarboxylase antibodies; HbA1c, glycated haemoglobin; IAA, Insulin auto-antibodies; MMTT, Mixed-Meal Tolerance Test; T1D, type 1 diabetes.

and placebo dosages, and identical labelling will be used for the ATG infusion bag and placebo.

### Trial participants, study design and oversight

There will be continuous recruitment of 114 participants, divided into seven sequential cohorts with variable dose allocation.

The first cohort of 30 participants will be randomised to 5 intervention arms, namely placebo, 2.5 mg/kg, 1.5 mg/kg, 0.5 mg/kg and 0.1 mg/kg ATG total dose in a 1:1:1:1:1 allocation ratio stratified by two age groups 12–17 and 18–25. Recruitment will initially be limited to 12–25 year olds until the first 10 participants 12–17 years of age to receive a non-placebo dose of ATG have been observed for 4 weeks after last dose administration. For the age step-down, a dose determining committee (DDC) will examine the safety data in this initial group of 10 non-placebo dose participants 12–17 years of age and will determine whether it is safe to recruit younger participants (aged 5–11 years) into the trial. The independent data monitoring committee (IDMC) will review the DDC decision for the safety age step down analysis and will need to give final approval to allow the step down in age to occur.

The rationale for this step down is that ATG has not yet been used in this indication and at these doses in paediatric participants aged 5–11 years. New-onset T1D participants aged 12–45 were already treated with 2.5 mg/kg ATG in a previous clinical trial by Haller et al reported in 2018 and 2019 and and no concerning safety signals were seen in the 1 year trial with 5-year follow up.[25 26 29]

Once the age step-down has occurred, randomisation into cohort 1 will continue until 30 participants have been randomised, in a 1:1:1:1:1 allocation ratio within three age stratification groups (5–9, 10–17 and 18–25). After cohort 1 has been recruited, recruitment to cohort 2 will begin, including stratification to the three age groups (5–9, 10–17 and 18–25).

Cohort 2 will consist of 12 participants randomised to 4 intervention arms, placebo, ATG 2.5 mg/kg and two middle ATG doses, in a 1:1:1:1 allocation ratio stratified by the three age groups. The choice of the middle doses will be made by the DDC using all available clinical, mechanistic and safety data from the previous cohort's data. Further a Bayesian model's predictions, using all MMTT C-peptide measurements in cohort 1 up to 12 months follow-up, will be used to help recommend cohort 2's middle dose selections to the DDC. CD4 + and CD8+ T cell counts will also be used for the DDC's decision because as shown by Haller et al[26] they represent an effectivity indicator for dose action: If these levels do not change then that dose will not be allowed to be taken forward regardless of the recommendation of the Bayesian model. The DDC will be required to come to a complete consensus for the middle dose(s) allocated to the next cohort. After cohort 2 has been recruited, recruitment to cohort 3 will begin.

Cohort 3 will consist of 12 participants randomised to 4 intervention arms, placebo, 2.5 mg/kg ATG and two middle doses decided by the DDC, in a 1:1:1:1 allocation ratio stratified by the three age groups. The DDC will be convened to determine the two middle doses to be applied to cohort 3 following the same procedure as described above, using all MMTT C-peptide measurements for all participants in all previous cohorts up to 12 months follow-up. The middle doses selected for cohort 3, or indeed any future cohorts, may be different to those selected for cohort 2, or previous cohorts. After cohort 3 has been recruited, recruitment to cohort 4 will begin.

Cohort 4 will consist of 15 participants randomised to 3 intervention arms, placebo, 2.5 mg/kg ATG and a single selected middle dose decided by the DDC, in a 1:1:1 allocation ratio stratified by the three age groups. The DDC will again be convened to determine the one middle dose to be used in cohort 4, following the same procedure as described above.

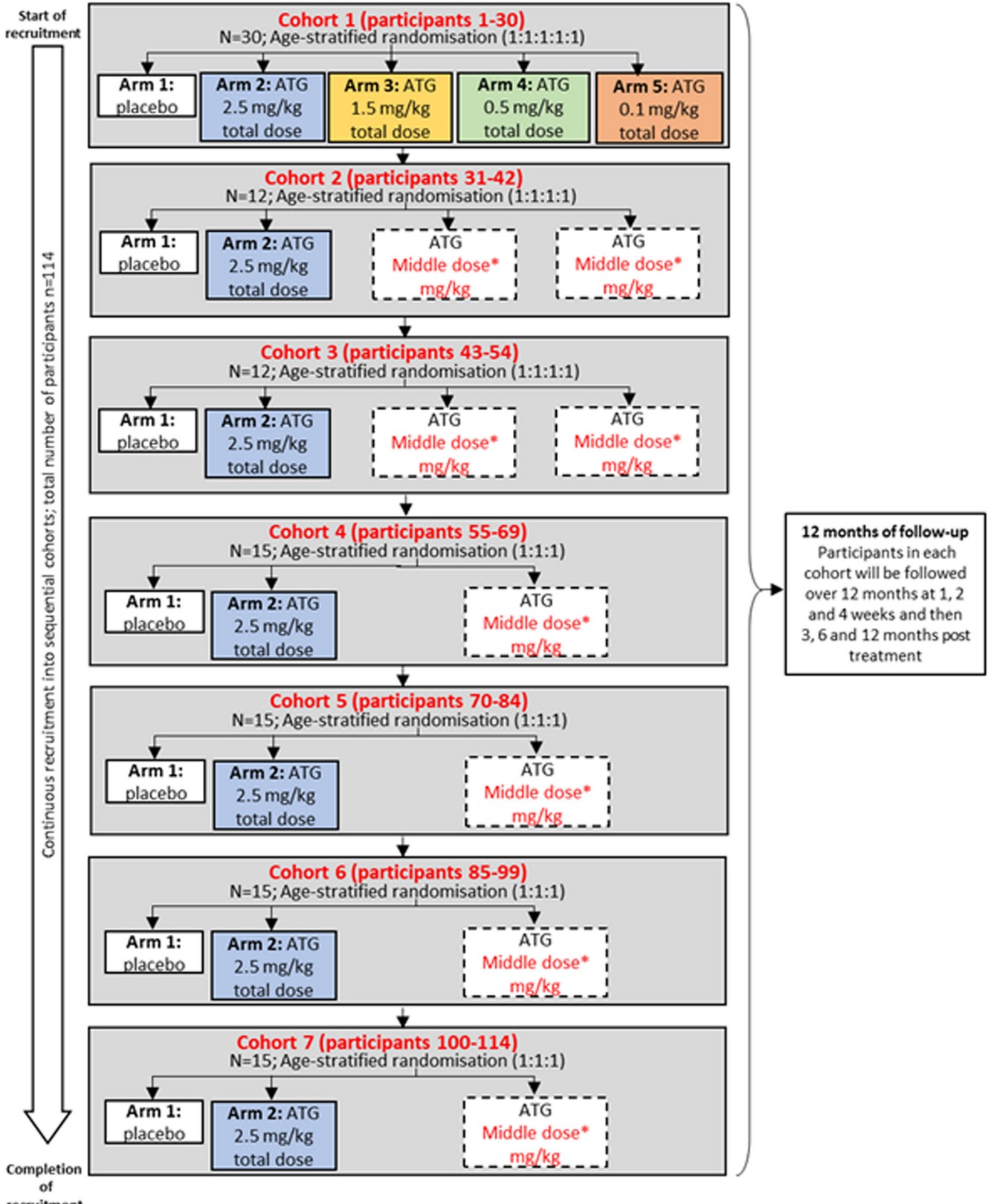

**Figure 1** MELD-ATG design. ATG total dose (mg/kg) will be divided into ttwo infusions on wo consecutive days. For safety, recruitment in cohort 1 will start in 12–25 years old before stepping down to <12 years old. *Middle dose(s) will be adjusted individually by cohort by the DDC and IDMC for cohorts 2–7 following review of all toxicity and early efficacy data of the preceding cohort(s). Middle dose(s) selected for the next cohort may differ from those selected for previous cohorts. ATG, antithymocyte globulin; DDC, Dose Determining Committee; IDMC, independent data monitoring committee; MELD, minimal effective low dose.

Cohorts 5, 6 and 7 will each consist of 15 participants randomised to three intervention arms, placebo, 2.5 mg/kg and selected middle doses, in a 1:1:1 allocation ratio within each age group. The above-described process will continue for each cohort up to Cohort 7 with additional individual interim analyses to identify the middle doses to be used in cohorts 5–7.

As randomisation is stratified by the three age groups and as it is only balanced within each strata, once the step down in age has been achieved, there is a potential for some imbalance between intervention arms on the completion of each cohort dependent on the age distribution of the participants. According to the study power calculation, 32 participants on the placebo and 32 participants on 2.5 mg/kg ATG dose are required to achieve the desired power, therefore the DDC will ensure that enough participants are allocated to these two groups while trying to balance the three age groups in the stratified randomisation, as part of each DDC meeting convened after each cohort's recruitment is completed. If there is an imbalance due to the age stratification within the randomisation the DDC will pick 2.5 mg/kg or placebo as the middle dose to ensure the power of the study; as we need to ensure that 32 participants are recruited to both the placebo and 2.5 mg/kg arm. This design allows sequential selection of the middle doses to be explored following review of all safety and early efficacy data by the IDMC and DDC to seek the minimum effective dose.

Potential participants will be identified by the clinical care teams at participating sites and will be approached by healthcare professionals and/or local research teams soon after T1D diagnosis. Potential eligible individuals or their parent/legal guardians (in case of children) will be provided with a verbal explanation of the study and applicable written participant information sheets (PISs). PISs will be translated into in the local language for the participating country. Once they have been given sufficient time to consider their participation in the trial, the participants/parents will be asked to provide written informed consent/assent to the study. Participants that withdraw from the trial after randomisation will not be replaced.

## Trial steering committee

Overall supervision of the trial rests with the trial steering committee (TSC). The TSC constitutes in accordance with current guidance. In particular, 66% of the TSC members are required in order to achieve quorum; 75% of members voting must be independent; the TSC chair is independent. A participant representative serves as a member of the TSC. The TSC aims to meet in person or by teleconference with a minimum frequency of 1/year (more frequent meetings may be required at the discretion of the TSC chair) as fully detailed in the TSC Charter. The TSC considers recommendations from the IDMC. The TSC will decide whether to modify the trial, or to seek additional data.

## IDMC and DDC

The IDMC charter details the purpose of this committee and that of the DDC, including: the description of the membership, terms of reference, roles, responsibilities, authority, decision-making and relationships of the IDMC and the DDC for this trial. The charter includes the timing of meetings, methods of providing information to and from the IDMC and the DDC, frequency and format of meetings, statistical issues, and relationships with other committees. Briefly, it is planned that six interim analyses are undertaken during the MELD-ATG trial and in light of these interim analyses and safety endpoints, the IDMC will advise the TSC of its recommendations regarding trial modification, continuation or termination of the trial. The IDMC and DDC charter expands on the above.

## Trial management group

The trial management group (TMG) comprise investigators and individuals closely involved in running of the trial. The TMG aims to meet more frequently than the TSC to ensure that all practical details of the trial are progressing well.

## Inclusion and exclusion criteria

Box 1 lists the study's inclusion and exclusion criteria. Potential participants may not enter the trial if any of the exclusion criteria listed in box 1 apply.

## Informed consent

The informed consent form (ICF) has been approved by the national or local ethical committees and is in compliance with Good Clinical Practice (GCP), local regulatory requirements and legal requirements. The investigator or design must ensure that each trial participant, or their legally acceptable representatives, are fully informed about the nature and objectives of the trial and possible risks associated with their participation. Countries will follow national requirements for consent of children and adults. The investigator or design will obtain written ICF each participant, or the participant's legally acceptable representatives if the participant is under the age of consent, before any trial-specific activity is performed. Participants under the age of consent will be asked to assent to trial participation.

The ICF/assent forms and any change made to them during the course of this trial, must be prospectively approved by the Ethics committee, following any other local regulatory or legal requirements. The investigator will retain the original of each participant's signed ICF. Copies of the signed PIS/ICF will also be provided to the participant or parents/legal guardians. As with PISs, ICFs will be translated into the local language for the participating country. Should a participant be unable to adequately read the documents, and therefore require a verbal translation of the trial documentation by a locally approved interpreter/translator, it is the responsibility of the individual investigator to use locally approved translators and to ensure a record of the process is present

## Box 1 Eligibility criteria

Inclusion criteria

1. Has given written informed consent to participate; or have a parent or legal guardian provide written informed consent. Individual under the age of consent will be asked to assent to trial participation.
2. Be aged ≥5 years to ≤25 years at written informed consent/assent.
3. Have been diagnosed with type 1 diabetes (T1D) within 3–9 weeks of planned treatment day 1.
4. Have random C-peptide levels ≥200 pmol/L measured at screening, as tested centrally.
5. Have one or more diabetes-related autoantibody (GADA, IA-2A or ZnT8A) present at screening, as tested centrally.
6. Will be ≥6 weeks from last live immunisation at planned treatment day 1 and be willing to forgo live vaccines during the trial until 6 months post-treatment.
7. Be willing to comply with intensive diabetes management.

Exclusion criteria

1. Type 2 Diabetes.
2. Evidence of prior or current tuberculosis infection.
3. Clinically significant abnormal full blood count, renal function or liver function at screening, including: (A) Immunodeficient or clinically significant chronic leucopaenia, neutropaenia, lymphopaenia or thrombocytopaenia at the screening visit, according to local reference ranges; (B) Evidence of liver dysfunction with aspartate aminotransferase or alanine transaminase greater than three times the upper limit of normal (ULN), at screening; (C) Evidence of renal dysfunction with creatinine greater than 1.5 times the ULN at screening, adjusted for the age of the patient; (D) Clinically significant clotting disorder, according to local reference ranges.
4. Requiring use of other immunosuppressive or immunomodulation agents, including chronic use of systemic steroids.
5. Any active chronic infections at screening, or any active acute or chronic infections at baseline or on treatment day, which would contraindicate any additional immunosuppression
6. Seropositive for HIV, hepatitis B or hepatitis C infection at screening.
7. Positive for SARS-CoV-2 based on local testing regimen.
8. Unwilling to use appropriate contraception if sexually active during the trial, from date of written informed consent until completion of the 12-month follow-up visit.
9. Any history of malignancies.
10. Current or ongoing use of non-insulin pharmaceuticals that affect glycaemic control.
11. Active participation in another T1D treatment interventional trial in the previous 30 days prior to screening (excluding treatment with insulin).
12. Any prior treatment with antithymocyte globulin (ATG), Abatacept or anti-CD3 antibodies.
13. Known allergy to ATG or to similar products, or hypersensitivity to rabbit proteins or to any of the excipients.
14. Any condition, complicating medical issues or abnormal clinical laboratory results that the investigator judges may adversely affect trial conduct, cause increased risk to the participant or compromise the trial results.
15. Pregnant and breastfeeding women.

in source documentation. Any new information which becomes available, which might affect the participant's willingness to continue participating in the trial will be communicated to the participant in a timescale and via a medium deemed appropriate by the sponsor, depending on the level of risk posed by the new information. Record of the communication should be recorded in source documentation.

### Randomisation and blinding

Following written informed consent/assent, participants will be allocated a unique participant trial ID number and only deidentifiable information (partial date of birth and sex) will be collected. Participants will be randomised centrally using a web-based randomisation system (Sealed Envelope at https://www.sealedenvelope.com/).

Randomisation will be based on block randomisation within age strata: 5–9, 10–17 and 18–25 years. Five blocks will be used for cohort 1, to be generated by Sealed Envelope. Subsequently the modified randomisation lists provided by Cambridge Clinical Trials Unit (CCTU) will use four blocks for cohorts 2–3, and three blocks for cohorts 4–7. The allocation ratios are: 1:1:1:1:1 for cohort 1, 1:1:1:1 for cohorts 2 and 3 and 1:1:1 for cohorts 4–7 within each age strata. Treatment will be randomly assigned in each cohort sequentially as described above. Access to the web-based randomisation system will provided to designated individuals at participating sites, via individual user accounts, and instructions will be provided in the MELD-ATG Trial Manual.

Trial participants and research teams will be blinded to the treatment allocation for the duration of the trial. Site pharmacies are unblinded. Randomisation is blinded. A concealment list is provided to the pharmacist in order to unblind and prepare the appropriate infusion. The unblinded individuals keep the treatment information confidential and will not discuss or release information on treatment allocation. Pharmacies prepare the infusions of ATG and placebo so that they are similar in packaging and labelling.

In the event of a valid medical or safety event, the responsibility to break the treatment code resides solely with the investigator. The online randomisation system will be used for emergency unblinding. Appropriately trained and delegated site staff will be given the necessary access rights and permissions to access this facility. The name, contact details of the unblinder and reason for unblinding will be recorded within the system. The unblinder will not be shown the treatment allocation on-screen. Instead the allocation will be sent by email and should be printed and retained confidentially within the investigator side file (ISF). An email stating that an unblinding has taken place will be automatically sent to the coordination team for oversight purposes. Wherever possible, members of the research team should remain blinded. Participants whose treatment assignment has been unblinded before completion of trial treatment must be discontinued from trial intervention but should continue to be monitored in the study.

### Patient and public involvement statement

MELD-ATG uses the support of the patient advisory committee (PAC) from within INNODIA to ensure that

the views of patients and their families are incorporated within the study. The TSC has a dedicated member of the PAC who has been involved from the beginning of the design of the trial and will be supporting till completion. The importance of trust in all documentation put forward to participants and their families has been appreciated by the study team thereby having documentation reviewed by the PAC. The PAC has welcomed the MELD-ATG study as it sees a low-dose study as positive (as opposed to a high-dose study). The PAC wanted to highlight that the side effects of these drugs can be highly inconvenient (particularly for younger patients). Articulating the reasons for long durations of infusion of ATG was key in gaining the PAC buy in and ultimately allowing them to support the study in articulating benefits of the study to patients and their families. It is a great step forward in making sure that at the heart of research patients are able to give their opinions, be listened to and ultimately change the course of the treatment of their conditions.

## Trial procedures

The study procedures are reported in detail in table 2 and a trial flow chart is shown in figure 2. The trial duration for each trial participant will be approximately 13 months, consisting of a 2–3 week screening/baseline period, a 2-day treatment period and a 12-month follow-up period.

Highly effective contraceptive measures are to be used by women of childbearing potential (WOCBP) with male partner(s) for the duration of this trial. Female participants in MELD-ATG are considered WOCBP (ie, fertile) following menarche. Male participants with female partner(s) meeting the criteria of WOCBP are required to use adequate contraception for the duration of the trial. Furthermore, all male participants should refrain from donating sperm for the duration of the trial.

Coregistration in the INNODIA longitudinal study and MELD-ATG is not permitted; MELD-ATG participants may be offered participation in the INNODIA longitudinal study following failure to satisfy all eligibility criteria after screening assessment, or following completion of all assessments for the MELD-ATG 12 months follow-up visit. Unaffected family member participants in the INNODIA longitudinal study who subsequently receive a new diagnosis of T1D and started on insulin therapy may be approached for MELD-ATG participation. Co-registration in other research studies may be permitted providing they will not interfere with MELD-ATG outcomes and only following discussion with the trial coordination team.

Following informed consent/assent, the participant will be registered in the electronic Case Report Forms (eCRF) and a unique participant trial ID number generated. Registration in the eCRF should be performed on the same date as written informed consent/assent, or as soon as possible thereafter. All identifiable information such as full name, contact details and date of birth will be registered locally following local policies and regulations.

## Screening

The screening visit should be within 3±3 weeks from first insulin injection (as surrogate for T1D diagnosis), in order to allow sufficient time to complete screening assessments, confirm eligibility, complete and review all baseline assessments, randomise and start treatment day 1 within 6±3 weeks from T1D diagnosis.

Assessments performed at screening are as follows: demographics (age, sex, ethnicity), date of T1D diagnosis, vaccination history ≤6 weeks of planned treatment day 1, review of medications, medical history, including diabetes-related medical history, and AEs since written informed consent/assent.

In addition, blood samples should be collected for the following measures: autoantibodies (Glutamic acid decarboxylase antibodies-GADA, Insulin auto-antibodies-IAA, IA-2 antibodies-IA-2A or Zinc transporter 8 antibody-ZnT8A) to be tested centrally, C-peptide to be tested centrally, full blood count (FBC), renal and liver function, clotting studies Prothrombin time and Activated partial thromboplastin time (PT +APTT), Epstein Barr Virus (EBV) serology, HIV, SARS-CoV-2 and hepatitis B and C. Throughout the trial, a maximum of 3 mls blood/kg participant[32] will be taken at each visit, with no more than 100 mls per visit.

Following review of the screening assessments by the local medical team, including autoantibody and C-peptide results from central testing, participants will be declared eligible or ineligible for MELD-ATG. Participants who are ineligible will be informed of the results of the screening visit and the reason(s) for ineligibility explained. Ineligible participants may be offered to join the INNODIA longitudinal study (ClinicalTrials.gov identifier: NCT03936634) where appropriate. The reason for ineligibility will be recorded in the eCRF.

More information on MELD-ATG study samples, toxicology and normal care, MMTT and DBS measurements can be found in online supplemental information). Further, table 3 shows individual study sample storage and analysis methods split by their study purpose.

## Baseline assessments for those who are eligible and have consented

The baseline visit will be performed in eligible participants within ≤3 weeks from screening visit and 1–7 days prior to treatment day 1 to collect baseline data and samples, and to further assess their suitability to receive trial treatment. Participants will arrive having fasted for 8 hours. Water is allowed.

The following baseline data will be collected for all participants: daily insulin regimen at time of baseline visit, HbA1c result within ±1 week of T1D diagnosis (if available), concurrent medications and vaccinations (any changes since screening visit), relevant family medical history and AEs since the screening visit.

The following baseline assessments should be conducted for all participants: physical examination (height and weight); general physical exam (blood

**Table 2** Schematic representation of assessments at study visits

| Visit | Screening | Baseline | Treatment | | FU V1 | FU V2 | FU V3 | FU V4 | FU V5 | FU V6 |
|---|---|---|---|---|---|---|---|---|---|---|
| | | | Day 1 | Day 2 | | | | | | |
| Timeline | 3±3 weeks from T1D diagnosis | ≤3 weeks after screening visit; 1–7 days before treatment; Fasted | Day 1 6±3 weeks from T1D diagnosis | | 1 week (±1 day); Post end of trial treatment | 2 weeks (±1 day) | 4 weeks (±1 day) | 3 months (±3 weeks); Fasted | 6 months (±3 weeks); Fasted | 12 months (±3 weeks); Fasted |
| **Assessments** | | | | | | | | | | |
| Consent/assent/registration | x‡‡ | | | | | | | | | |
| Eligibility criteria | x | | | | | | | | | |
| Demographics* | x | | | | | | | | | |
| Medical history | x | | | | | | | | | |
| Medications & vaccinations | x | x | | | x | x | x | x | x | x |
| AEs† | x | x | x | x | x | x | x | x | x | x |
| HIV, Hep B, Hep C, EBV | x | x‡ | | | | | | | | |
| FBC, renal and liver function | x | x | x | x | | x | x | x | | |
| PT and APTT§ | x | | x§ | x§ | | x§ | x§ | x§ | | |
| HbA1c¶ | | x | | | | | | x | x | x |
| Insulin regimen | | x | | | x | | | x | x | x |
| Family medical history | | x | | | | | x | x | x | x |
| Randomisation | | x | | | | | | | | |
| Physical exam** | | x | x | x | x | x | | x | x | x |
| Vital signs†† | | x | x | x | x | x | | x | x | x |
| MMTT | | x | | | | | | x | x | x |
| Urine pregnancy test‡‡ | | x | | | | x | x | x | x | x |
| SARS-CoV-2 test | | x | | | | | | | | |
| ATG/placebo administration | | | x | x | | | | | | |
| **Research samples** | | | | | | | | | | |
| Autoantibodies | x | | | | | | | | | x |
| Random plasma C-peptide | x | | | | | | | | | |
| Diabetes-related genotyping | | x | | | | | | | | |
| CD4/CD8 ratio | | x | | | x | x | x | x | x | x |
| Exploratory studies (Omics)§§ | | x | | | x | x | x | x | x | x |
| Urine and stools for Omics¶¶ | | x | | | | | | x | x | x |
| Home | | | | | | | | | | |

Continued

**Table 2** Continued

| Visit | Screening | Baseline | Treatment | | FU V1 | FU V2 | FU V3 | FU V4 | FU V5 | FU V6 |
|---|---|---|---|---|---|---|---|---|---|---|
| | | ≤3 weeks after screening visit; 1–7 days before treatment | Day 1 | Day 2 | 1 week | 2 weeks | 4 weeks | 3 months | 6 months | 12 months |
| | | | | | (±1 day) | (±1 day) | (±1 day) | (±3 weeks) | (±3 weeks) | (±3 weeks) |
| | | Fasted | | | | | | Fasted | Fasted | Fasted |
| **Timeline** | 3±3 weeks from T1D diagnosis | | Day 1 6±3 weeks from T1D diagnosis | | | Post end of trial treatment | | | | |
| DBS (monthly ±1 week) | | x*** | | | | x††† | x | x | x | x |
| CGM (14 days post visit) | | | | | | | | x | x | x |

*Demographics to include age, sex and ethnicity (where allowed).

†Recording of all AEs must start from the point of written informed consent/assent, regardless of whether a participant has yet received a medicinal product.

‡EBV PCR is required at baseline for participants who were EBV seronegative at screening. HIV, hepatitis B/C and EBV serology are not repeated at baseline.

§PT and APTT should be tested at screening for all participants but only at subsequent visits if screening results were abnormal (but not beyond exclusion criterion).

¶HbA1c does not need to be repeated at baseline or FU if a result is available from standard of care within 7 days (baseline visit) or 2 weeks (FU visits 4, 5 and 6).

**General physical exam; baseline to additionally include height and weight, menarcheal status (pre or post), pubertal staging (via self-assessment form if required), FU visits 4, 5 and 6 to additionally include height and weight.

††Blood pressure, heart rate, respiratory rate and temperature at all visits from baseline. Treatment days 1 and 2 to additionally include oxygen saturation.

‡‡Urine pregnancy tests for women of childbearing potential only.

§§Omics includes some or all of: transcriptomics, small/miRNA, genetics, lipidomics, proteomics, metabolomics, the microbiome and immunomics. Blood sample types required include whole blood, serum, plasma and PBMC at different visits. Refer to MELD-ATG Trial Manual for per-visit requirements.

¶¶Urine and stool samples may be collected ±1 week of the visit.

***Baseline DBS and blood glucose measurements to be performed in parallel with the baseline MMTT.

†††Follow up DBS and blood glucose measurements to be collected monthly at home, starting within the first 4 weeks after treatment.

‡‡‡Written informed consent/assent is to be taken before enrolment and registration in eCRF and any trial-specific assessments.

AEs, adverse events; APTT, Activated partial thromboplastin time; ATC, antithymocyte globulin; CGM, continues glucose monitoring; DBS, dried blood spot; EBV, Epstein-Barr virus; FBC, full blood count; FU, follow-up; HbA1c, glycated haemoglobin; MELD, minimal effective low dose; MMTT, Mixed Meal Tolerance Test; ;PT, Prothrombin time.

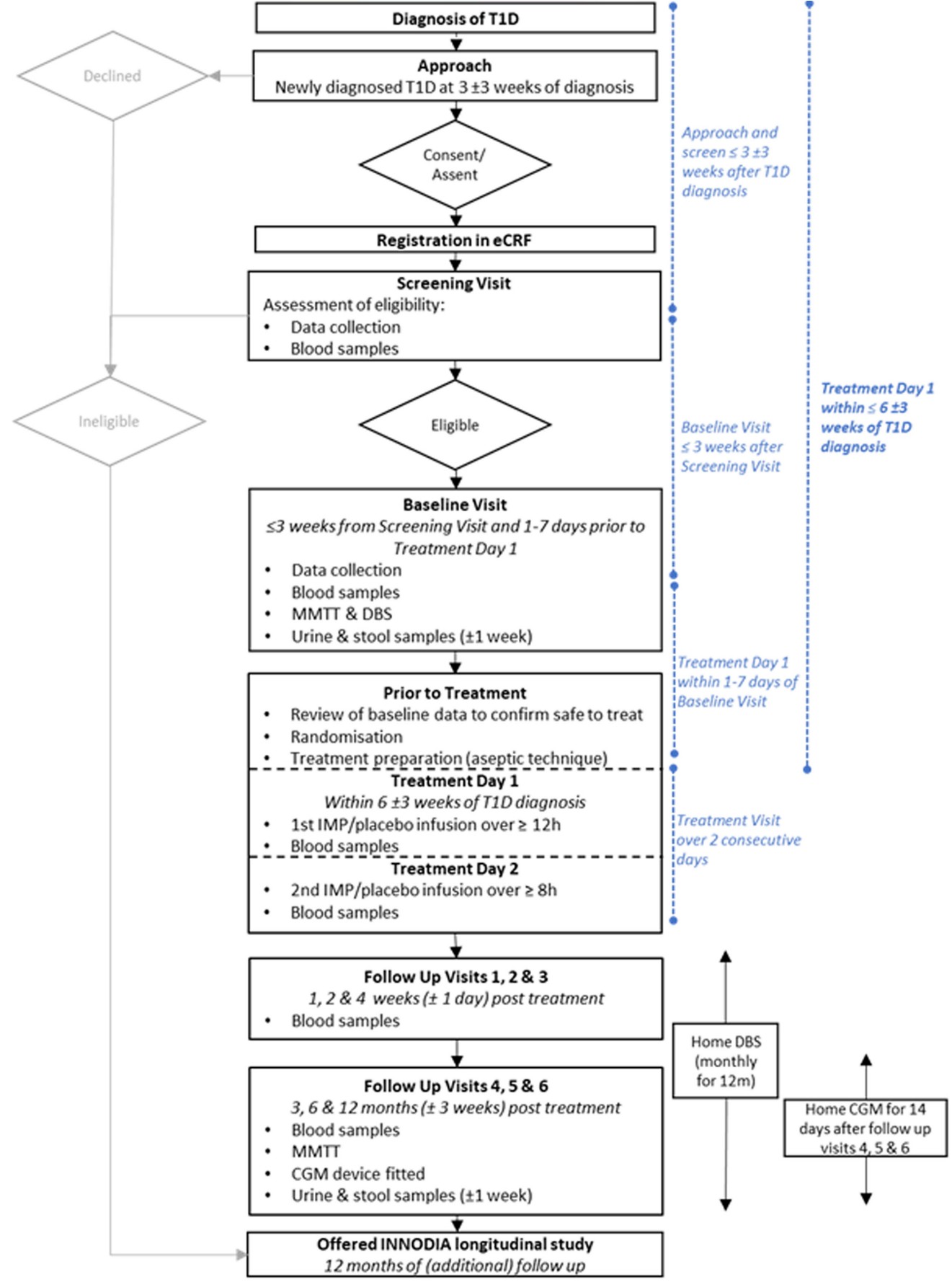

**Figure 2** Trial flow chart. CGM, continuous glucose monitoring; DBS, dry blood spot; MMTT, Mixed Meal Tolerance Test; T1D, type 1 diabetes. eCRF, electronic Case Report Forms; IMP, Investigational medicinal product

**Table 3** Study sample storage and analysis methods

| Sample For | Sample | Before analysis | Analysis | Storage after analysis |
|---|---|---|---|---|
| Screening eligibility criteria | Random plasma C-peptide | Plasma collected for C-peptide analysis will be sent on dry ice to University of Cambridge | CBAL | Samples will be kept at ≤69°C in freezers with temperature monitoring by University of Cambridge |
| | Diabetes-related autoantibodies (GADA, IAA, IA-2A or ZnT8A) | Serum will be sent for analysis fresh within 24 hours to the PEDIA Laboratory (Helsinki, Finland) | PEDIA Laboratory | Samples will be kept at ≤69°C in freezers with temperature monitoring by the PEDIA Laboratory |
| Primary outcome | AUC stimulated C-peptide over first 2 hours of MMTT at 12 months follow-up | Plasma collected for C-peptide analysis will be sent on dry ice to the University of Cambridge, where they will be stored until analysis at <-69°C in freezers with temperature monitoring | Analysis in batches CBAL | Before and after analysis, samples will be kept at ≤69°C in freezers with temperature monitoring by University of Cambridge |
| Secondary outcome | AUC stimulated C-peptide over first 2 hours of MMTT at baseline, 3, 6 and 12 months follow-up | Plasma collected for C-peptide analysis will be sent on dry ice to the University of Cambridge, where they will be stored until analysis at ≤69°C in freezers with temperature monitoring | Analysis in batches by CBAL | Samples will be stored at ≤69°C in freezers with temperature monitoring by University of Cambridge |
| | DBS C-peptide at observed times | DBS cards collected for C-peptide analysis will be stored locally at participating sites (frozen ≤69°C) until shipment on dry ice to the University of Cambridge | Analysis by CBAL | After analysis by CBAL, samples will be stored by the University of Cambridge |
| | CD4/CD8 ratio over 12 months | Whole blood will be sent fresh following collection within 48 hours to an INNODIA immune hub or accredited local hospital laboratory | Analysis by INNODIA immune hubs or accredited local hospital laboratory for determination of CD4/CD8 ratios | No storage required, samples are consumed or remaining discarded after analysis. |
| | HbA1c (all time points) | Whole blood collected for HbA1c measurement will be sent to local hospital accredited laboratory routinely performing this analysis as standard of care | Analysis by local hospital accredited laboratory. HbA1c results will be entered into the eCRF by participating sites. | Samples will be discarded after analysis as per normal local protocols |
| | T1D-associated autoantibodies at baseline and 12 months | Serum will be sent for analysis fresh within 24 hours to the PEDIA Laboratory | Analysis by the PEDIA Laboratory | After analysis samples will be kept at ≤69°C in freezers with temperature monitoring by the INNODIA central laboratory by the PEDIA Laboratory |
| Exploratory studies | Biomarkers related to immunological changes and β-cell death/ survival | Whole blood will be sent fresh following collection to an INNODIA immune hub | Fresh blood immune assays and PBMC isolation performed by INNODIA immune hubs | Isolated PBMCs will be stored in liquid nitrogen by an INNODIA immune hubs |
| | Diabetes-related genotyping | Blood cells collected for genotyping will be stored locally at participating sites (frozen ≤69°C) until shipment on dry ice to the University of Cambridge where they will be further forwarded to the genotyping lab for DNA extraction and analysis | JDRF/Wellcome Trust Diabetes and Inflammation Laboratory | Remaining cells and extracted DNA samples will be stored at ≤69°C in freezers with temperature monitoring by JDRF/ Wellcome Trust Diabetes and Inflammation Laboratory |

Continued

| Table 3 | Continued | | | |
| --- | --- | --- | --- | --- |
| Sample For | Sample | Before analysis | Analysis | Storage after analysis |
| CBAL, Core Biochemical Assay Laboratory; DBS, dry blood spot; HbA1c, mixed meal tolerance test; JDRF, Juvenile Diabetes Research Foundation; MMTT, Mixed Meal Tolerance Test; PBMC, Peripheral blood mononuclear cell; PEDIA, Pediatric Diabetes Research Group ; T1D, type 1 diabetes. | | | | |

pressure, heart rate, temperature, respiratory rate); blood samples for HbA1C- (if not available from standard of care testing within 7 days of baseline visit); CD4+/CD8 +T cell count ratio; biofluids collection for exploratory mechanistic omics studies; MMTT-including blood samples for measurement of glucose and C-peptide; DBS to be collected during the MMTT (>5 years of age only); urine samples (±1 week); stool samples (±1 week); and SARS-CoV-2 test for current infection based on local testing regimen.

Additional baseline assessments for applicable participants only include physical exam (premenarche or postmenarche; pubertal staging-via self-assessment form if required); urine pregnancy test for WOCBP and EBV PCR for participants who are EBV seronegative at screening.

### Randomisation

Prior to randomisation and treatment, the investigator or design will review baseline data to reconfirm that the participant is safe to receive trial treatment and continues to fulfil the eligibility criteria. Randomisation should be performed before the participant is due to receive their first trial treatment infusion, allowing sufficient time for treatment preparation. Randomisation should only occur for whom the eligibility criteria can be confirmed.

### Treatment day 1 and 2

Participants will be under strict medical supervision in hospital for the treatment visit. The total dose of trial treatment will be given over two consecutive days by intravenous infusion.

Treatment day 1 will be ≤6 ±3 weeks from T1D diagnosis. Changes to health and medications since the baseline visit should be checked including a SARS-CoV-2 test, and blood pressure, heart rate, oxygen saturation, respiratory rate and temperature will be recorded. The participant should also be physically examined for evidence of current infection prior to administration of premedication. It is recommended that concomitant medication with heparin and hydrocortisone, and premedication with corticosteroid, antihistamine and acetaminophen, is given as described in the MELD-ATG Pharmacy Manual, to limit ARs from peripheral ATG/placebo infusion. Blood glucose levels may be affected by premedication and will be managed by the responsible investigator.

The first part-dose of 0.5 mg/kg or 0.1 mg/kg ATG, or placebo of the same volume, will be given as an intravenous infusion over a minimum of 12 hours. AEs throughout the infusion until the end of the visit will be recorded. At the end of the infusion, blood will be collected for FBC, clotting studies (PT and APTT), if PT and/or APTT were abnormal but not beyond exclusion criteria at the screening visit, as well as renal and liver function. The FBC will be reviewed in order to confirm that it is safe to continue with the second treatment infusion.

The second treatment infusion will be given no sooner than 12 hours and no later than 24 hours after the end of the first infusion, over a minimum of 8 hours. As on day 1, vital signs (including blood pressure, heart rate, oxygen saturation, respiratory rate and temperature) will be recorded before administration premedication. Any changes to health and non-trial (excluding insulin) medications during treatment day 1 should also be checked. It is recommended that participants receive premedication as on treatment day 1, as described in the MELD-ATG Pharmacy Manual.

### Monitoring post-treatment before discharge

During the infusion, vital signs (including blood pressure, heart rate, oxygen saturation, respiratory rate and temperature) will be recorded every 30 min for the first 2 hours, followed by every 60 min or as indicated for clinical signs or symptoms. AEs throughout the infusion until the end of the visit will be recorded as they develop. Guidance on management of the infusion site, CRS, allergic reaction, serum sickness and haematological effects is provided below. At the end of the infusion, blood will be collected for FBC, clotting studies (PT and APTT), if PT and/or APTT were abnormal but not beyond exclusion criteria at the screening visit, as well as renal and liver function.

### Subsequent assessments: follow-up visits 1–6

The following assessments are performed at every follow-up visit: general physical exam, blood pressure, heart rate, temperature, respiratory rate, diabetes care (including insulin regimen and diabetic ketoacidosis), changes in medications and vaccinations, changes in AEs, a SARS-CoV-2 test and urine pregnancy test for WOCBP (follow-up visits 2–6 only). Blood samples will be collected for CD4+/CD8 +T cell count ratio determination and exploratory studies (Omics). Follow-up visit 1 will take place 1-week post end of treatment (±1 day) and the recurring assessments listed above will be carried out.

Follow-up visit 2 will take place 2 weeks post end of treatment (±1 day) while follow-up visit 3 will take place 4 weeks post end of treatment (±1 day). In addition to the aforementioned recurring assessments, safety blood samples will be collected for FBC, clotting studies (PT and APTT), if PT and/or APTT were abnormal but not

beyond exclusion criteria at the screening visit, as well as well as renal and liver function for safety monitoring. There will be regular contact between participants and parent/legal guardians and the research team between 2 and 4 weeks post-treatment to monitor AEs, particularly development of serum sickness.

At follow-up visits 4, 5 and 6 (3, 6 and 12 months post-treatment), participants will have a 120 min MMTT during which blood will be collected for measuring glucose and C-peptide. For MMTT measurements please see details in online supplemental appendix below.

Participants will be fitted with a CGM device at follow-up visits 4, 5 and 6 (3, 6 and 12 months post-treatment) and requested to collect CGM measurements for 14 days after each visit. CGM devices (CE marked) will be provided by DexCom Inc. (USA) for use in MELD-ATG and training will be provided to the participant/responsible adult prior to first use. For monthly DBS samples taken at home please see details in the online supplemental appendix below.

Follow-up visit 4 will take place 3 months post end of treatment (±3 weeks). CGM will be collected by the participant for 14 days post visit. Participants will be advised to arrive having fasted for 8 hours. Water is allowed. In addition to the aforementioned recurring assessments, height and weight, changes in family medical history, safety blood samples for FBC, clotting studies (PT and APTT), if PT and/or APTT were abnormal but not beyond exclusion criteria at the screening visit, as well as renal and liver function for safety monitoring will be collected. In addition, research blood samples will be collected for HbA1c (if a result is not available from standard of care testing within ±2 weeks of the visit), along with MMTT as described above, as well as urine (±1 week) and stool samples (±1 week).

Follow-up visit 5 will take place 6 months post end of treatment (±3 weeks). CGM will be collected by the participant for 14 days post visit. Participants will be advised to arrive having fasted for 8 hours. Water is allowed. In addition to the aforementioned recurring assessments, height and weight, and changes in family medical history. In addition, research blood samples will be collected for HbA1c (if a result is not available from standard of care testing within ±2 weeks of the visit), along with MMTT as described above, as well as urine (±1 week) and stool samples (±1 week).

Finally, follow-up visit 6 will take place 12 months post end of treatment (±3 weeks). CGM will be collected by the participant for 14 days post visit. Participants will be advised to arrive having fasted for 8 hours. Water is allowed. In addition to the aforementioned recurring assessments, height and weight, and changes in family medical history. In addition, research blood samples will be collected for HbA1c (if a result is not available from standard of care testing within ±2 weeks of the visit) and T1D-associated autoantibodies (GADA, IAA, IA-2A and ZnT8A) along with MMTT as described above, as well as urine (±1 week) and stool samples (±1 week).

## End of trial participation

The end of trial participation is defined as completion of the final follow-up visit. Once complete, the CGM device will be returned to the participating site (may be offered to participants but only if support costs are available locally). Participants enrolled into this trial already receive the appropriate standard of care for T1D, and this care continues after the trial. Following completion of MELD-ATG participation, participants will be offered to join the INNODIA longitudinal study (ClinicalTrials.gov identifier: NCT03936634; www.innodia.eu). Participation in this further study is optional and would involve a separate PIS and ICF/assent form. Participation would allow for data to be collected on the MELD-ATG participants for an additional 12 months delivering 24 months outcome data. Consent would be sought to link MELD-ATG trial data with INNODIA longitudinal study data. Written informed consent will have been sought on joining the trial to confirm that participants, or their appropriate legal representative, give permission to be contacted to inform them of trial results, potential future follow-up, and/or future interventional and/or observational T1D studies organised by the INNODIA consortium.

## Early discontinuation/withdrawal of participants

A participant may choose to not start or to discontinue trial treatment early at any time before or during trial treatment or to withdraw from future participation in the trial at any time. This may be done without necessarily giving a reason, and without any personal disadvantage and without affecting their usual clinical care. An investigator may choose to discontinue trial treatment early (eg, due to toxicities) or to stop the participation of a participant after consideration of the benefit/risk ratio. Possible reasons are: (1) continued non-compliance with the trial protocol, (2) technical grounds (eg, participant moves) and (3) early termination at the request of the CI/PI. A reason must be provided for investigator decisions.

Discontinuation of trial treatment (participant or investigator decision) does not automatically entail withdrawal from the trial. Participants who discontinue trial treatment early are expected to continue with the follow-up assessments and sample collection per protocol, as part of the intention-to-treat analysis. Discontinuation from trial follow-up visits should be the last possible solution. Participants that withdraw after randomisation will not be replaced. Trial treatment discontinuation or participant withdrawal and the reason (where available) will be documented in the eCRF. In cases of withdrawal, permission to retain samples and data already collected will be documented in the eCRF.

## ATG preparation, dose and administration

Trial treatment infusions (intravenous) will take place over two consecutive days. ATG (Thymoglobuine) used in this trial must be stored, handled, prepared and dispensed as detailed in the current SmPC.[22] This includes the requirement for administering Investigational medicinal

product (IMP) and observing the participant in a hospital setting under medical supervision. Medical personnel and equipment must be readily at hand to provide emergency treatment if necessary, including in case of anaphylaxis. The diluent must be 0.9% sodium chloride; neither dextrose nor glucose solutions are permitted for this T1D trial population. Concomitant medication with hydrocortisone, heparin, and premedication with corticosteroid, antihistamine and paracetamol/acetaminophen is recommended, according to local procedures for ATG infusion to minimise the risk of expected ARs.

All participants of the trial will be under strict medical supervision for the duration of each infusion and carefully monitored before, during and after the infusions. Participants will be discharged no sooner than 2 hours after finishing the trial treatment infusions, or according to the investigator's clinical judgement. In the event of early trial treatment discontinuation, or if the maximum of 60 hours has been reached without completing the second infusion, trial treatment can be discontinued without gradual tapering of the dose as per.[22]

### Placebo
Placebo will be an identical volume of sodium chloride 0.9%. Premedication requirements for placebo treatment are the same as for active IMP to maintain the blind for participants and the investigator/nurse (concomitant medication and premedication can affect blood glucose). Placebo will come from local hospital stock. Pharmacies will be responsible for applying the trial-specific labelling to placebo (and ATG) infusion bags and participant-specific details will be added to the labelling prior to dispensing. More information on the legal status, accountability and supply of the IMP/Placebo can be found in online supplemental information.

### Known drug reactions and interaction with other therapies
In new-onset T1D subjects, no concerning safety signals were seen in low dose (2.5 mg/kg) ATG-treated subjects (12–45 years) in a 1-year trial with 5-year follow-up, as reported by Haller et al.[25 26 29] Predominant AEs reported include serum sickness, CD4 lymphocyte decrease, CRS, fever, influenza-like symptoms and rash. Serum sickness was observed in the majority of participants (72.4%) who received low dose ATG, but this was predictable and short-lived. In addition, none of the participants required extended hospitalisation or readmission due to CRS or serum sickness, and no cases of grade 4 serum sickness or CRS were reported.

In MELD-ATG, premedication described in the MELD-ATG Pharmacy Manual will be given to all participants to minimise the risk of these side effects, and participants will be closely monitored. None of the participants who received low dose ATG in this previous trial developed a serious infection. Infection screening is part of the screening and baseline assessments for MELD-ATG, to ensure individuals with infections are not given ATG.

Management of serum sickness, CRS and haematological effects in MELD-ATG are provided below. The safety and effectiveness of ATG in new onset T1D paediatric participants <12 years needs confirmatory further study in controlled trials. However, for the approved indications (at a higher total dose than 2.5 mg/kg), data based on limited European studies and compassionate use in US indicates that the dose, adverse reaction (AR) profile and efficacy are not different than that reported in adolescents and adults. Known reactions to ATG in the licenced indications are described in the SmPC.[22]

No drug interaction studies with ATG have been reported[33]; however, certain groups of medications are contraindicated for reasons other than interaction with ATG. The following would interfere with interpretation of the trial and should be avoided throughout: (1) agents that influence insulin sensitivity or secretion (eg, pramlintide, sulfonylureas, metformin, diphenylhydantoin, thiazide or other potassium-depleting diuretics, β-adrenergic blockers, niacin), (2) any medication that may result in immunosuppression or immunomodulation and (3) systemic glucocorticoids (except when required as premedication for ATG/placebo administration or for the treatment of CRS or serum sickness). If participants receive, or if the investigator believes that participants must receive, any of these medications, it must be documented in the source data and eCRF, as per other concomitant medications. For the participant's safety, live vaccinations should be avoided from within ≥6 weeks prior to treatment day 1 until 6 months post-treatment, as stated in the eligibility criteria. Killed vaccines are permitted. Drugs not listed above will be also permitted as per investigators' discretion and should be listed in the eCRF.

### Advice on SARS-CoV-2/COVID-19 for the MELD-ATG trial
The MELD-ATG team are constantly revising our advice in a prepared COVID-19 bulletin on COVID-19 as the situation changes. The most up to date version of our COVID-19 bulletin can be accessed by contacting the MELD-ATG office (MELD-ATG@medschl.cam.ac.uk and/or MELD-ATG@uzleuven.be).

### Concomitant treatments
Concomitant medications will be assessed at each trial visit. Usual treatment with insulin, either via multiple injections or pump therapy, will be continued during the trial as advised by the investigator or other medical professional responsible for the participant's standard-of-care insulin therapy. Exceptions where adapted insulin regimens may be required are the MMTTs and DBS home collections. The use of non-insulin pharmaceuticals that affect glycaemic control will not be allowed for the trial duration.

### Compliance with trial treatment
Participants are treated under strict medical supervision, with the trial treatment being administered by healthcare

professionals. Compliance with treatment may be assessed by referring to medication administration records and any remaining unused ATG/placebo after treatment has been stopped, which should be returned to the site pharmacy for reconciliation.

## Dose delays and missed doses

Following review of the FBC obtained following completion of the first trial treatment (ATG or placebo) infusion, the dose of the second infusion may be reduced by 50% in certain scenarios.

## Evaluation of AEs

Sponsor expects that AEs are recorded from the point of written informed consent/assent regardless of whether a participant has yet received a medicinal product. Individual AEs should be evaluated by the investigator. This includes the evaluation of its seriousness, and any relationship between the IMP(s) and/or concomitant therapy and the AE (causality). More information on the definitions for assessment of safety in MELD-ATG can be found in online supplemental information.

Assessment of seriousness defines whether the event is an AE, serious AE (SAE) or an Serious adverse reaction (SAR). Assessment of causality is broken up into the following categories: (1) Definitely: A causal relationship is clinically/biologically certain. This is therefore an AR; (2) Probable: A causal relationship is clinically/biologically highly plausible and there is a plausible time sequence between onset of the AE and administration of the IMP and there is a reasonable response on withdrawal. This is therefore an AR; (3) Possible: A causal relationship is clinically/biologically plausible and there is a plausible time sequence between onset of the AE and administration of the IMP. This is, therefore, an AR (4) Unlikely: A causal relation is improbable and another documented cause of the AE is most plausible. This is therefore an AE; (5) Unrelated: A causal relationship can be definitely excluded and another documented cause of the AE is most plausible. This is therefore an AE. Unlikely and unrelated causalities are considered not to be IMP related. Definitely, probable and possible causalities are considered to be IMP related. A pre-existing condition must not be recorded as an AE or reported as an SAE unless the condition worsens during the trial and meets the criteria for reporting or recording in the appropriate section of the eCRF.

All events should be graded for severity according to the NCI-CTCAE Toxicity Criteria (V.5.0).

AEs and ARs should be recorded in the medical notes and the appropriate section of the eCRF. SAEs and SARs should be reported to the Sponsor and IMP manufacturer as detailed below.

## Expected AEs/SAE

The following are (S)AEs that could be reasonably expected for this trial population during the course of the trial. These events are exempt from being reported as

SAEs and will be recorded in the eCRF only: (1) diabetic ketoacidosis, (2) hospital admissions related to diabetes (planned and unplanned) and (3) hypoglycaemic event which requires assistance from a third party.

The following AEs are known side effects of participating in this trial. They are generally not serious in nature and will not be recorded in the eCRF: (1) injection site and mild skin reactions following IV administration of ATG/placebo, (2) mild bruising at venipuncture site, (3) mild reaction to plasters and (4) vasovagal response during venipuncture.

## Reporting SAEs

Each principal investigator (PI) must record all AEs and report SAEs in English to the Clinical trial Coordinating centre (CTCC) in Cambridge, using the trial-specific SAE form, within 24 hours of their awareness of the event; the sponsor must then immediately forward the form to Sanofi. The co-ordinating centre will perform an initial check of the report, request any additional information, and will pass it on to the chief investigator (CI) or nominated clinician without delay.

The CI is responsible for the onward notification of all SAEs to the sponsor immediately but not more than 24 hours of first notification. A detailed record of all SAE's reported to the sponsor will be kept. The CI is also responsible for prompt reporting of all SAE findings to the competent authority if they could: (1) adversely affect the health of participants, (2) impact on the conduct of the trial, (3) alter the risk to benefit ratio of the trial, (4) alter the competent authority's authorisation to continue the trial in accordance with Directive 2001/20/EC. PIs will be informed of all suspected unexpected serious adverse reactions (SUSARs) and will ensure it is reported to their relevant authorities in accordance with their own regulations.

SAEs will also be reported to the IMP manufacturer in accordance with the pharmacovigilance agreement and be reviewed at the next IDMC meeting.

## Reporting of suspected unexpected serious adverse reactions

All suspected ARs related to an IMP (the tested IMP and comparators) which occur in the concerned trial, and that are both unexpected and serious (SUSARs) are subject to expedited reporting. Each PI must notify all SUSARs, in English, to the CTCC in Cambridge, using the trial-specific SAE form, immediately as they become aware of the event.

The CI must report all the relevant safety information previously described, to the: (1) sponsor, (2) competent authorities in the concerned member states, (3) Ethics Committee in the concerned member states and (4) IMP manufacturer. The CI shall inform all investigators concerned of relevant information about SUSARs that could adversely affect the safety of participants.

For fatal or life-threatening SUSARs, all parties listed in the paragraph above must be notified as soon as possible but no later than seven calendar days after the trial team

and sponsor has first knowledge of the minimum criteria for expedited reporting. In each case, relevant follow-up information should be sought and a report completed as soon as possible. It should be communicated to all parties within an additional eight calendar days. Non-fatal and non-life-threatening SUSARs must be reported as soon as possible but no later than 15 calendar days after first knowledge of the minimum criteria for expedited reporting. Further relevant follow-up information should be given as soon as possible.

Information on the final description and evaluation of an AR report may not be available within the required time frames for reporting. For regulatory purposes, initial expedited reports of SUSARs should be submitted within the time limits as soon as the minimum following criteria are met: (1) a suspected IMP, (2) an identifiable participant (eg, trial participant code number), (3) an AE assessed as serious and unexpected, and for which there is a reasonable suspected causal relationship, (4) an identifiable reporting source and (5) when available and applicable a unique clinical trial identification (EudraCT number) and a unique case identification (ie, sponsor's case identification number). In case of incomplete information at the time of initial reporting, all the appropriate information for an adequate analysis of causality should be actively sought from the reporter or other available sources. Further available relevant information should be reported as follow-up reports. In certain cases, it may be appropriate to conduct follow-up of the long-term outcome of a particular reaction. Electronic reporting is the expected method for expedited reporting of SUSARs to the competent authority. The format and content as defined by the competent authority should be adhered to. SUSARs will be reported in English to the IMP manufacturer in the format as described in the agreement.

### Pregnancy reporting

Pregnancy occurring from written informed consent/assent until the end of the trial period, that is, 12 months following the end of the treatment period should be reported and followed up until the outcome of the pregnancy is known. All pregnancies within the trial (either the trial participant or the participant's partner) should be reported to the CI and the sponsor and IMP manufacturer using the relevant Pregnancy Reporting Form within 24 hours of notification. Pregnancy is not considered an AE unless a negative or consequential outcome is recorded for the mother or child/fetus. If the outcome meets the serious criteria, this would be considered an SAE.

### Toxicity management

All dose interruptions/reductions, and the reasons for the interruptions/reductions, are to be recorded in the eCRF. Medications given to manage toxicities are also to be recorded in the eCRF. In the event of toxicity reactions (hypersensitivity, allergic reaction, serum sickness and haematological effects) during the trial treatment infusion, the guidance in the below sections may be followed regarding dose interruptions and discontinuation. Any rescue medication required administered should be in line with advice given to local investigators and in appropriate facilities. In the event of toxicity reactions after the end of trial treatment infusion, any rescue medication required should be administered in line with local procedures.

CRS is caused by a large, rapid release of cytokines into the blood from immune cells affected by immunotherapy. Signs and symptoms of CRS include fever, nausea, headache, rash, rapid heartbeat, low blood pressure and breathing difficulties. Most participants are expected to have a mild reaction, but sometimes, the reaction may be severe or life threatening. Participants may experience CRS following ATG/placebo infusion. If CRS severity is mild (grade 1) then the trial treatment can be continued. If the CRS severity is moderate (grade 2) then the trial treatment may be interrupted and restarted once signs and symptoms improve. If a subsequent dose of ATG/placebo further exacerbates the signs and symptoms of CRS despite following the guidelines in the MELD-ATG Trial Manual, the treatment must be permanently discontinued, and additional supportive/resuscitative measures given as clinically indicated. However, if CRS severity is severe (≥grade 3), then trial treatment should be permanently discontinued.

In rare cases, participants may experience hypersensitivity, which refers to immediate allergic, IgE-mediated reactions to ATG. Such participants primarily develop skin rash and respiratory distress early in the course of the infusion (usually within the first hour). For such reactions, the infusion should be discontinued, and the participant managed according to local hospital procedures.

Other allergic reactions to trial treatment infusion may be managed based on the severity of the allergic reaction. A mild (≤grade 1–2) allergic reaction will allow trial medication to be restarted at the discretion of the investigator. With either a severe (grade 3) reaction or a life-threatening (grade 4) allergic reaction, trial medication should be permanently discontinued. In case of anaphylaxis (begins at CTCAE grade 3 or higher), the trial treatment infusion should be terminated immediately, and appropriate emergency treatment should be initiated. Equipment for emergency therapy for anaphylactic shock must be readily available.

Serum sickness from host immunisation against rabbit protein may occur 7–15 days after the first dose of ATG. It is a reaction that is similar to an allergy. Signs and symptoms can take as long as 14 days after exposure to appear, and may include signs and symptoms commonly associated with hypersensitivity or infections including rashes, itching, arthralgia (especially finger and toe joints), fever (as high as 40°C and usually appears before rash), lymphadenopathy (particularly near the site of injection, head and neck), malaise, hypotension, splenomegaly, glomerulonephritis, protein or blood in the urine or shock. The participants than may require glucocorticoid treatment

for supportive care. The dose will depend on the severity of signs and symptoms and guidance is provided in the MELD-ATG Trial Manual. The least amount of steroid required to provide symptomatic relief of serum sickness, due to the effect of glucocorticoid on blood glucose.

Haematological effects such as thrombocytopaenia and/or leucopaenia (including lymphopaenia and neutropaenia) may occur and are reversible. FBC will be reviewed following completion of the first ATG/placebo infusion. A 50% dose reduction is permissible for the second ATG/placebo infusion if required as per the SmPC[22] if the platelet count is between 50 and 75 x10ˆ9/L (50 000 and 75 000 cells/mm$^3$) or if the white cell count (WCC) is between 2 and 3 x10ˆ9/L (2000 and 3000 cells/mm$^3$). Before this happens, the FBC will first be repeated and if the platelet count is over 75 x10ˆ9/L (75 000 cells/mm$^3$) on repeat or if the WCC is over 3 x10ˆ9/L (3000 cells/mm$^3$), then the second ATG or placebo infusion will be given as planned. If on repeat FBC, the platelet count is still between 50 and 75 x10ˆ9/L (50 000 and 75 000 cells/mm$^3$) or if the WCC is still between 2 and 3 x10ˆ9/L (2000 and 3000 cells/mm$^3$), then the second ATG or placebo infusion dose will be reduced by 50%. Specifically, dose reduction will be achieved by administering 50% of the infusion volume (250 mL) instead of the full volume (500 mL). Dose reduction, and the reason, will be documented in the eCRF. Discontinuation of trial treatment (ATG/placebo) should be considered if persistent and severe thrombocytopaenia (<50 x10ˆ9/L or 50 000 cells/mm$^3$ for platelets) occurs or leucopaenia (< 2 x10ˆ9/L or 2000 cells/mm$^3$ for WCC count) develops on repeat FBC.

### Infusion site management

Infusion site reactions are relatively common with both IMP and placebo but can be minimised by application of a cold pack to the infusion site 2–3 min before and immediately after the infusion to reduce the associated discomfort.

### Statistics overview

The trial design consists of seven cohorts, each recruited sequentially, with between three and five treatment arms (there will be fewer arms for later cohorts). The total sample size is 114 participants allocated to treatment arms. The primary endpoint of interest is the area under the C-peptide curve (AUC) over the first 2 hours (using all available time points) obtained after an MMTT (AUC C-peptide) at 12 months follow-up; it will be undergo the transformation ln(x+1) for the primary analyses.

All model assumptions will be checked via graphical means such as plotting residuals versus dose and fitted values. We will transform our primary outcome, MMTT AUC C-peptide value to ln(AUC C-peptide +1). The transformed MMTT AUC C-peptide value is assumed to be normally distributed; this distributional assumption will be assessed using a Q-Q plot and whether the residuals of the model deviate from a straight line. If the residuals are

not normally distributed, then the outcome will be transformed to improve this assumption. If no transformation is available, then non-parametric methods will be used.

### Evaluation of results (definitions and response/evaluation of outcome measures)

The primary assessment of efficacy is carried out at the completion of the trial (once all patients for cohort 7 have completed the 12 months follow-up or the last patient last visit) for the 2.5 mg/kg dose of ATG and using the primary outcome of area under the stimulated C-peptide response curve over the first 2 hours of the MMTT at 12 months follow-up. We will transform the MMTT AUC C-peptide value to ln(AUC C-peptide +1). The primary analysis will be a statistical significance test.

### Statistical methods for primary analyses

The primary endpoint, the area under the C-peptide curve over the first 2 hours (using all available measurements within the first 2 hours) of an MMTT (AUC C-peptide) at 12 months (after transformation) will be analysed using a linear regression model at the end of the trial. The model will have fixed effects of time and categorical dose along with an interaction term between time and dose with participant as a random effect and the baseline measurement of MMTT AUC C-peptide (transformed using the ln(x+1) function). The primary contrast of interest is the mean difference in transformed MMTT AUC C-peptide between dose level 2.5 mg/kg and placebo with a 95% CI. The statistical test will be the Wald test or t-test on the 2.5 mg/kg parameter in the model and significance is assessed at the 5% level. If there is a significant effect for the 2.5 mg/kg versus placebo mean difference, then every other dose level will be compared with the placebo dose at the 5% level using t-tests. If there is not a significant effect for the 2.5 mg/kg dose, then no other tests will be carried out. This forms a closed testing procedure for the primary endpoint and the family wise error rate is controlled at 5%. The lowest dose below 2.5 mg/kg that is found to be statistically significantly different to placebo will be declared the minimally effective low dose (MELD).

### Statistical methods for secondary analyses

The secondary endpoints measured repeatedly over time will be analysed via a mixed effects model. Within the model time and categorical dose will be fixed effects along with an interaction term between time and dose with participant as a random effect. If required, the models may include additional covariates which may be potential factors that are confounding the relationship between dose and outcomes. The main additional covariate will be age of diagnosis. Subgroup analyses will be considered for a select list of potential covariates, the subgroup treatment effect will be analysed using an interaction test and additional factors will be included in the model to conduct this test. As an extra sensitivity analysis, we will assume a linear effect of dose and rerun the model

with that assumption. A detailed statistical analysis plan will be produced before the final database lock.

## Interim analyses

There will be six interim analyses, one after each of the six cohorts to determine which doses should be administered to each subsequent cohort of participants using a Bayesian model using all MMTT C-peptide measurements in all previous cohorts up to 12 months follow-up. Every cohort will have a placebo arm and a 2.5 mg/kg arm. Recruitment to the next cohort will not be halted and the DDC will be convened to determine the middle dose or doses (in the case of cohorts 2 and 3) to be applied to the next cohort. The trial statistician will produce a report with predictions from the Bayesian model to help recommend the next cohort's middle dose selections for cohorts 2–7. The DDC will be required to come to a complete consensus for the middle dose(s) allocated to the next cohort combining clinical opinion, safety outcome information from the previous cohort's data as well as the Bayesian model recommendations to make their decision. They will also examine CD4 + and CD8+T cell counts by dose to ensure that the dose is having some effect. If these levels do not change then that dose will not be allowed to be taken forward regardless of the recommendation of the Bayesian model. All safety outcomes by age will also be reviewed by the DDC as part of these interim analyses. The DDC and IDMC charter will expand on the above.

Specifically, the interim analyses will include an exploratory n sample decision to drop a dose/doses based on the CD4+/CD8 +T cell count ratio outcome at 1, 2, 4 and 12 weeks using a repeated measures model with fixed effects of dose and nominal days along with an interaction term between time and dose since randomisation. The mean differences between each dose level and placebo will be estimated with a 95% CI. CIs that indicate no change in CD4+/CD8 +T cell count ratio from baseline (ie, include 0) may indicate that this dose may be inactive and dropped from further cohorts.

These results will be used in conjunction with the C-peptide modelling which will be a Bayesian model of ln(AUC C-peptide +1). Only vague priors will be used in the model. The model will include fixed effects for categorical time and dose level along with an interaction term between time and dose and a random effect for participant. This model will be used to calculate the predictive probability that the mean ln(AUC C-peptide +1) is different from placebo for each dose level (including 2.5 mg/kg) given that the remaining participants to be allocated to the middle dose are given this dose level. If no such dose level is found then the next cohort of participants will be allocated to the 2.5 mg/kg dose level (improving power). Other stopping boundaries are not considered because the interim analyses are based on a mixture of incomplete follow-ups and fully observed follow-ups and subsequent analyses may suggest a middle dose not equal to 2.5 mg/

kg. There is no sequential hypothesis testing done on the primary analysis.

## Number of participants to be enrolled: sample size calculation

The SD used for this calculation for the AUC C-peptide endpoint after an MMTT over 2 hours at 12 months was 0.27 nmol/L/min as per[34]; this SD was also estimated as 0.264 nmol/L/min using a data extract from INNODIA from February 2 2020, using the transformed AUC C-peptide value and restricting the extract population to those 5–25 years old. With this SD, 90% power, and 5% significance level, then 32 participants per arm will be needed to detect a change of 0.22 nmol/L/min comparing the 2.5 mg/kg arm to placebo on the transformed ln(AUC C-peptide +1) scale using a two sided two-sample t-test power calculation. Then if this test was significant, there would be 80% power if there were 30 participants (the exact number is unknown but is likely to be fewer than on the 2.5 mg/kg arm) on each middle dose (not 2.5 mg/kg) to detect a change of 0.2 nmol/L/min when comparing to the placebo arm. The sequential testing, that is, a closed testing procedure, means that the familywise error rate of 5% is controlled.

Using the INNODIA mean value of AUC C-peptide, from 175 participants in the longitudinal study aged 5–25 years old, of 0.77 nmol/L/min we would expect to observe a mean difference of 0.42 nmol/L/min between 2.5 mg/kg and the placebo group, while on the transformed scale we would be observe 0.22 nmol/L/min. All tests are superiority tests and the tests are two-sided tests.

## Criteria for the premature termination of the trial

There are no statistical conditions for premature termination of the trial. The IDMC may recommend to the TSC early termination of the trial due to, for example, clear benefit or harm of a dosage, clear lack of benefit, safety concerns, slow recruitment or external evidence. If the study is prematurely terminated or suspended, the sponsor/coordination team will promptly inform the investigators and the regulatory authority(ies) of the reason(s) for termination or suspension. The ethical committee(s) also will be promptly informed by the sponsor/investigator and provided with the reason(s) for the termination or suspension, as specified by the applicable regulatory requirement(s).

## Procedure to account for missing or spurious data

For all key endpoints including the area under the stimulated C-peptide response curve over the first 2 hours of an MMTT at 12 months post-treatment, CD4 +T cells and CD8 +T cells and fasting/stimulated DBS C-peptide measurements a repeated measures mixed effects model will be used as the analysis allowing outcomes with missing data to be incorporated. All participants who are randomised will be included in this analysis and the model will have fixed effects for time and dose along with an interaction term between time and dose and participant is a random effect. All available repeated measurements

of MMTT AUC C-peptide in the same individual over time will be included in the analysis and an unstructured autocorrelation will be estimated for each dose level if the data allow. These estimates assume that the missing data are missing at random. If the missing data is non-ignorable then a sensitivity analysis will be performed. The secondary endpoints will also be analysed using a mixed effects model similar to the one described above.

### Definition of the end of the trial
The trial ends when the last participant has completed the trial. Translational endpoints will be investigated, and the data cleaned for the primary analysis and reported within 6 months from end of trial.

### Data management and eCRF
All data will be collected in the eCRF, which will be pseudo-anonymised. The eCRF will be accessed via a web interface and allows live (immediate) entry during the participant's trial visit. All entries are encrypted at both ends and are fully auditable (date/time user stamp). Trial data in the eCRF must be extracted from and be consistent with the relevant source documents. The eCRF and data queries must be completed in a timely manner; all entries are automatically recorded and auditable. It remains the responsibility of the local PI for the timing, completeness and accuracy of the eCRF. The eCRF will be accessible to trial coordinators, data managers, the investigators, clinical trial monitors, auditors and inspectors as required. Central data management will be performed by the University of Cambridge with eCRFs built by the University of Copenhagen, Denmark. Data will be collected in English using MELD-ATG eCRFs. Data entry is the responsibility of delegated individuals at the participating site. Please refer to the MELD-ATG Trial Manual for full information.

### Source data
To enable peer review, monitoring, audit and/or inspection the investigator must agree to keep records of all participants (sufficient information to link records) and all original signed ICFs. Sites are required to have source data (paper or electronic). This includes but is not limited to participant medical records (paper or electronic), notes made during trial visit (paper or electronic; signed and dated), results of clinical tests for example, local laboratory blood test results (FBC, LFT, renal function, HbA1C, glucose, HIV/EBV/hepatitis, SARS-CoV-2), approach and Consent Logs, signed ICF/Assent Forms, prescriptions and pharmacy accountability/destruction logs, participant diary, completed self-assessment forms for pubertal staging, if used and dried blood spot collection forms. The eCRF can be considered source data for research sample tracking purposes (but not sample results). Individual participant medical information obtained as a result of this trial is considered confidential and disclosure to third parties is prohibited.

### Data protection and participant confidentiality
All investigators and trial site staff involved in this trial will comply with the requirements of the EU General Data Protection Regulations and the Data Protection Act 2018 and local institution policy with regards to the collection, storage, processing, transfer and disclosure of personal information and will uphold the Act's core principles. In particular, the investigator and site staff must ensure that no participant identifiable information (including name, address, and hospital number) is transmitted to the trial team or sponsor. Every participant will be allocated a unique trial ID number that will link all of the clinical information held about them on the trial database and only deidentifiable information (partial date of birth and sex) will be collected. It will also be used in all correspondence with participating clinical trial sites. At no point in presentations or publications of trial data will individual participants be identified.

### Protocol compliance and breaches of GCP
Prospective, planned deviations or waivers to the protocol are not allowed and must not be used. For example, it is not acceptable to enrol a participant if they do not meet one or more eligibility criteria or restrictions specified in the trial protocol. In the event that eligibility criteria need to be changed/amended then they must first be approved by regulatory authority/ethical committee/ other relevant authorities via a substantial protocol amendment before they can be implemented. Protocol deviations, non-compliances, or breaches are departures from the approved protocol. They can happen at any time but are not planned. They must be adequately documented on the relevant forms and reported to the CI and sponsor immediately. Deviations from the protocol which are found to occur constantly again and again will not be accepted and will require immediate action and could potentially be classified as a serious breach. Any potential/suspected serious breaches of GCP must be reported immediately to the sponsor without any delay.

### Monitoring, trial documentation and archiving
The investigator must make all trial documentation and related records available should a regulatory inspection occur. Should a monitoring visit or audit be requested, the investigator must make the trial documentation and source data available to the sponsor's representative. All participant data must be handled and treated confidentially. The sponsor's monitoring frequency will be determined by an initial risk assessment performed prior to the start of the trial. A detailed monitoring plan will be generated detailing the frequency and scope of the monitoring for the trial. Throughout the course of the trial, the risk assessment will be reviewed, and the monitoring frequency adjusted, as necessary.

Each participating site is responsible for archiving their own trial data (including source data, ISF, PSF) for the appropriate time period as determined by the regulations governing clinical trials in place at the time of archival.

The archiving facility may be at the participating site or at another appropriate location off-site as per local policy. The trial team will advise when the site may commence archiving. The site will need to provide the name and address of the archival facility to the Trial team. In case of audit or inspection following archival, the participating site will be expected to retrieve the relevant documentation within a reasonable time frame.

## ETHICS AND DISSEMINATION
### Ethical committee review
Before the implementation of substantial amendments in a participating country, we will obtain approval of the trial protocol, protocol amendments, ICFs and other relevant documents, for example, advertisements and GP information letters if applicable from the applicable ethical committee(s). All correspondence with the ethical committees(s) will be retained in the Trial Master File (TMF)/ISF. Update reports will be submitted to ethical committee(s) in accordance with committee requirements.

### Regulatory compliance
The trial will not commence in a country until a Clinical Trial Authorisation (or equivalent country-specific authorisation) is obtained from the national regulatory authority or any other relevant agency. The protocol and trial conduct will comply with the European Clinical Trials Directives 2001/20/EC and 2005/28/EC and the relevant local national legislation (including for Belgium the law on experiments dated 7 May 2004) and any relevant amendments. Development safety update reports (DSURs) will be submitted to the national regulatory authority or any other relevant agency in other participating countries in accordance with national requirements. DSURs will be prepared in English by the University of Cambridge with supervision from the sponsor. DSURs will be submitted to the regulatory authorities by participating countries, in accordance with national requirements. Periodic progress reports will be submitted to the local ethics committees/other relevant authorities by participating countries, in accordance with national requirements. Participating countries are responsible for ensuring reports adhere to local/national requirements.

### Protocol amendments
Protocol amendments must be reviewed and agreement received from the sponsor for all proposed substantial amendments prior to submission to the national regulatory authority, ethical committee and any other relevant agency/ethical committee. The only circumstance in which an amendment may be initiated prior to ethical and regulatory approvals is where the change is necessary to eliminate apparent, immediate risks to the participants (urgent safety measures). In this case, accrual of new participants will be halted until the national regulatory authority, ethical committee, or any other relevant

agency, approval has been obtained. All protocol amendments will be prepared (in English) by the University of Cambridge.

### Peer review
The protocol has been reviewed by the INNODIA Consortium Management Group, the INNODIA Consortium PAC, Cambridge University Hospitals National Health Service (NHS) Foundation Trust and Universitair Ziekenhuis Leuven. Additionally, the protocol has been reviewed by the IMP manufacturer Sanofi.

### Declaration of Helsinki and GCP
The trial will be performed in accordance with the spirit and the letter of the Declaration of Helsinki, the conditions and principles of GCP, the protocol and applicable local regulatory requirements and laws.

### GCP training
All trial staff must hold evidence of appropriate GCP training or undergo GCP training prior to undertaking any responsibilities on this trial. This training should be updated every 2 years or in accordance with each institution's procedures.

### Authorisation of participating sites
Prior to initiating a participating site, the following documentation is required: (1) PI and other key trial team staff CV (signed and dated) and GCP certificate, (2) ethical committee approval, (3) regulatory authority approval, (4) all relevant national and local institutional approvals, (5) participating Site Agreement executed, (6) a copy of all participant documentation (PISs, ICFs and Assent Form, etc) on local headed paper with local contact details added, (7) protocol signed and dated by PI, (8) delegation of Authority Log, (9) confirmation of randomisation system training, (10) confirmation of eCRF training, (11) example of IMP/placebo prescription, (12) confirmation of local pharmacy green light and (13) local laboratory accreditation (or equivalent) and reference ranges for the protocol-specified parameters.

When all the regulatory paperwork is in place, prior to site opening, an initiation visit will take place, either face-to-face or via a teleconference. This will be led by the INNODIA clinical trial accreditation team or delegate with as many of the local team present as is practicable. This initiation meeting constitutes training for the trial and it is therefore imperative that all members of the trial team who will be involved in the trial are represented at the meeting. A log of attendees will be completed during the meeting. The presentation slides will be provided to the site in advance of the meeting. A trial initiation form will be completed for each site initiation visit. Copies of all initiation documentation must be retained in the ISF and TMF. The sponsor's regulatory green light procedure will be followed. Following the green light, the initial supply of ATG will be ordered for shipment to the site on the authorisation of the coordinating centre coordinator. Following confirmation of receipt of the IMP at site, the

site will be opened for recruitment and the randomisation system opened to that site.

The PI has overall responsibility for the conduct of the trial at the participating site. In particular, the PI has responsibilities which include (but are not limited to): (1) ensuring the appropriate approvals are sought and obtained, (2) continuing oversight of the trial, (3) ensuring the trial is conducted according to the protocol, (4) ensuring consent is received in accordance with the protocol and national requirements, (5) ensuring that the ISF is accurately maintained, (6) delegation of activities to appropriately trained staff (this must be documented on the Delegation of Authority Log), (7) providing protocol or specialised training to new members of the trial team and ensuring that if tasks are delegated, the member of staff is appropriately trained and qualified, (8) appropriate attendance at the initiation meeting, (9) ensuring appropriate attendance at the TMG/TSC/IDMC/DDC meetings if required and ensuring appropriate safety information is made available to the coordinating centre team in advance of the meetings, (10) dissemination of important safety or trial-related information to all stakeholders at the participating site, (11) safety reporting within the timelines and assessment of causality and expectedness of all SAEs, (12) notification of all SAE's (including SUSARs) to their relevant authorities in accordance with their own regulations, (13) submission of annual reports in accordance with their own regulations, (14) facilitation of on-site and remote monitoring for the trial, including completion of remote monitoring reports as requested by the sponsor or their delegate and (15) archiving of the ISF on confirmation of post end of trial by the sponsor.

### Financial, insurance and publication policy

Each participating site to the clinical trial will accept full financial liability for harm caused to participants in the clinical trial caused through the fault or negligence of its employees, agents, subcontractors and honorary contract holders. Sponsor will be liable towards and arrange for insurance of participants in the clinical trial in accordance with local national legislation. Participant's reasonable expenses (eg, travel, accommodation) incurred as a direct result of taking part in the trial will be reimbursed by the participating site following local procedures. The publication policy is set in the INNODIA grant agreement.

### TRIAL STATUS

The trial is currently recruiting and enrolling participants. Date of First Enrolment was on 24 November 2020 in Leuven Belgium. MELD-ATG received regulatory and ethical approvals in Belgium from the Federal Agency for Medicines and Health Products in Saint-Gilles, Brussels, as of 10 September 2020. Initial regulatory approval has also been granted by the Paul-Ehrlich-Institut in Germany and the Medicines and Healthcare products Regulatory Agency in the UK as of February 2021. Ethical approval

has been granted in Hannoversche Kinderheilanstalt Auf der Bult (Hannover, Germany), University Medical Centre Ljubljana (Ljubljana, Slovenia) and Cambridge University Hospitals NHS Trust in the UK. Universite Libre de Bruxelles (Brussels, Belgium) is now also open to recruitment as of March 2021.

### DATA AVAILABILITY
#### Underlying data
No data are associated with this article.

#### Extended data
This project contains the following extended data:
1. Participant information.pdf (UK PIS, ICF and assent forms for all age groups).
2. Online supplemental information-Protocol appendix. pdf.

### REPORTING GUIDELINES
Standard Protocol Items: Recommendations for Interventional Trials checklist for 'MELD-ATG: phase II, dose ranging, efficacy study of ATG within 6 weeks of diagnosis of T1D'.

**Author affiliations**
[1]Centre for Trials Research, College of Biomedical and Life Sciences, Cardiff University, Cardiff, UK
[2]Department of Paediatrics, University of Cambridge, Cambridge, UK
[3]Cambridge Clinical Trials Unit, Cambridge University Hospitals NHS Foundation Trust, Cambridge, UK
[4]Pharmacy, Cambridge University Hospitals NHS Foundation Trust, Cambridge, UK
[5]Katholieke Universiteit Leuven/ Universitaire Ziekenhuizen, Leuven, Belgium
[6]Center for Protein Research, Kobenhavns Universitet Sundhedsvidenskabelige Fakultet, Kobenhavn, Denmark
[7]Diabetes Centre for Children and Adolescents, Children's Hospital Auf der Bult, Hannover, Germany
[8]Research Program for Clinical and Molecular Metabolism, University of Helsinki Faculty of Medicine, Helsinki, Finland
[9]Pediatric Research Centre, University of Helsinki Children's Hospital, Helsinki, Finland
[10]Department of Immunobiology, King's College London, London, UK
[11]INNODIA Patient Advisory Committee, Madrid, Spain
[12]INNODIA Patient Advisory Committee, Paris, France
[13]Department of Pediatrics, University of Florida, Gainesville, Florida, USA
[14]Sanofi-Aventis Deutschland GmbH, Frankfurt, Germany
[15]Wellcome Trust-MRC Institute of Metabolic Science, Cambridge University, Cambridge, UK

**Contributors** Writing-original draft preparation: CSW-B. Conceptualisation: DD, MJH, AM, CM, AN and AMS. Funding acquisition: DD, AM, CM, AN and AMS. Methodology: SiB, DD, AM and CSW-B. Writing-review and editing: OA, BA-F, SBr, SBo, AC, PJC, DD, KG, MJH, AEJH, MK, AM, MLM, CM, SEM, HM, AN, LO, JP, DP, AMS, TT, CSW-B and MW. Project administration: OA, BA-F, SBr, SBo, AC, PJC, DD, KG, AEJH, MK, AM, MLM, CM, SEM, HM, AN, LO, JP, DP, AMS, TT, CSW-B and MW.

**Funding** This clinical trial is funded by Innovative Medicine Initiative 2 Joint Undertaking (IMI2 JU) INNODIA under grant agreement no 115797. This Joint Undertaking receives support from the Union's Horizon 2020 research and innovation program and 'EFPIA', 'JDRF' and 'The Leona M. and Harry B. Helmsley Charitable Trust'. The IMP is supplied by INNODIA member Sanofi. The CE-marked CGM devices are provided by DexCom (USA). supported by IMI2-JU under grant agreement No 115797 (INNODIA) and No 945268 (INNODIA HARVEST). DD was supported by the UK NIHR Cambridge Biomedical Research Centre.The sponsor is Universitaire Ziekenhuizen Leuven (UZL), Leuven, Belgium. Sponsor Number:

S63466. Central Coordination is done on behalf of the sponsor by the University of Cambridge, UK (MELD-ATG@medschl.cam.ac.uk). The EudraCT Number is 2019-003265-17.

**Disclaimer** Sanofi—one funder of the Study—had a role in study design, decision to publish and preparation of the manuscript. The other study funders had no role in study design, data collection and analysis, decision to publish, or preparation of the manuscript.

**Competing interests** CM serves or has served on the advisory panel for Novo Nordisk, Sanofi, Merck Sharp and Dohme, Eli Lilly and Company, Novartis, AstraZeneca, Boehringer Ingelheim, Roche, Medtronic, ActoBio Therapeutics, Pfizer, Insulet and Zealand Pharma. Financial compensation for these activities has been received by KU Leuven; KU Leuven has received research support for CM from Medtronic, Novo Nordisk, Sanofi and ActoBio Therapeutics; CM serves or has served on the speakers bureau for Novo Nordisk, Sanofi, Eli Lilly and Company, Boehringer Ingelheim, Astra Zeneca and Novartis. Financial compensation for these activities has been received by KU Leuven. MJH is a scientific board member of SAbBiotherapeutics and has received research funding from Sanofi. AN and AMS are Sanofi employees and may hold shares and/or stock options in the company.

**Patient consent for publication** Not applicable.

**Provenance and peer review** Not commissioned; externally peer reviewed.

**ORCID iDs**
Charlotte S Wilhelm-Benartzi http://orcid.org/0000-0003-4927-6158
David Dunger http://orcid.org/0000-0002-2566-9304

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
