## [Reviewer comments · BMJ Open]

ARTICLE DETAILS

TITLE (PROVISIONAL)	Study Protocol Minimum effective low dose - Anti-human thymocyte globulin (MELD-ATG): Phase II, dose ranging, efficacy study of anti-thymocyte globulin (ATG) within 6 weeks of diagnosis of type 1 diabetes.
AUTHORS	Wilhelm-Benartzi, Charlotte; Miller, Sarah; Bruggraber, Sylvaine; Picton, Diane; Wilson, Mark; Gatley, Katrina; Chhabra, Anita; Marcovecchio, M. Loredana; Hendriks, A Emile; Morobé, Hilde; Chmura, Piotr; Bond, Simon; Aschemeier-Fuchs, Bärbel; Knip, M; Tree, Timothy; Overbergh, Lut; Pall, Jaivier; Arnaud, Olivier; Haller, MJ; Nitsche, Almut; Schulte, Anke; Mathieu, Chantal; Mander, Adrian; Dunger, Professor David

VERSION 1 – REVIEW

REVIEWER	Zhou, Zhiguang The Second Xiangya Hospital, Section of Endocrinology
REVIEW RETURNED	29-Aug-2021

GENERAL COMMENTS	Overall, the design of this phase II clinical trial is very clear and clean. It is of great interest to know whether the anti-human thymocyte globulin (MELD-ATG) is efficacy in Type 1 diabetes subjects. I have no more comments and suggestions for the study protocol. Because it is already very clear and detailed. The group and sample size are reasonable. The outcome is clear and right chosen. All the measurable lab test is sufficient and good to carry out. I just don't know if the journal would like to publish just a protocol like this to your readers?
---

REVIEWER	Leslie, Richard Queen Mary University of London, Blizard Institute
REVIEW RETURNED	08-Sep-2021

GENERAL COMMENTS	Immunotherapy has been used with some success in type 1 diabetes, One immunosuppressive agent that has shown promise in subjects aged 12-45 years is anti-thymocyte globulin (ATG), Thymoglobuline®, and further exploration in lower age groups and at lower doses is warranted. The study MELD-ATG is a phase 2, multi-centre, randomised, double-blind, placebo- controlled, multi-arm parallel-group trial in participants (n=114) 5-25 years diagnosed with T1D within 3– 9 weeks of planned treatment day 1. Participants are being randomised to placebo, 2.5 mg/kg, 1.5 mg/kg, 0.5 mg/kg and 0.1 mg/kg ATG total dose in a 1:1:1:1:1 allocation ratio. The next six cohorts of 12-15 participants will be randomised to placebo, 2.5 mg/kg, and one or two selected middle ATG total doses in a
--

	1:1:1:1 or 1:1:1 allocation ratio, as dependent on the number of middle doses, given intravenously over 2 consecutive days. The primary objective will be to determine the changes in stimulated C-peptide response over the first 2 hours of a mixed meal tolerance test (MMTT) at 12 months for 2.5mg/kg ATG arm versus the placebo. Conditional on finding a significant difference at 2.5 mg/kg, a minimally effective dose will be sought. Secondary objectives include the determination of the effects of a particular ATG treatment dose on 1) stimulated C-peptide, 2) HbA1c, 3) daily insulin dose, 4) time in range by intermittent continuous glucose monitoring (CGM) measures, 5) fasting and stimulated dry blood spot (DBS) C-peptide measurements. 1. Strengths The study protocol and the participants are well versed in these trials and the format has been extensively used under the INNODIA umbrella. The project objective is worthwhile, the aim to examine if younger subjects and lower doses can be used successfully are fully justified in the opinion of this reviewer. Inclusion and exclusion criteria are well established plus a position on live vaccines 2. Weakness It may not work. That aside the authors are conversant with the many issues such studies raise and have provided a protocol that is extensively tested in this area. 3. Tragically David Dunger recently died so they will need to alter the email. 4. Novel features in this study include the use of time in range and dried blood spot c-peptide and glucose. This exploration study is worthwhile but at a preliminary stage in development. 5. The reviewer notes with approval that exploratory mechanistic studies are to be performed.
--	---

REVIEWER	Elhenawy, Yasmine Ain Shams University, Pediatric and Adolescent Diabetes Unit (PADU), Department of Pediatrics, Faculty of Medicine, Ain Shams University, Egypt
REVIEW RETURNED	30-Oct-2021

GENERAL COMMENTS	MELD-ATG is a phase II, dose ranging, efficacy study of (ATG) in new onset T1D; including participants 5-25 years. The study is the first evaluating efficacy and safety of ATG among patients aged 5-11 years and the first as well trying to identify the minimally effective low dose of ATG (MELD). An important well designed interventional study that will provide possible data on the safety, tolerability, and efficacy of 2.5 mg/kg ATG and even lower doses in pediatric, adolescent and adult population with newly diagnosed T1D. Early Intervention with preservation of beta- cells of the pancreas represent a paradigm shift in managing T1D.
---

VERSION 1 – AUTHOR RESPONSE

Reviewer: 1

Prof. Zhiguang Zhou, The Second Xiangya Hospital

Comments to the Author:

Overall, the design of this phase II clinical trial is very clear and clean. It is of great interest to know whether the anti-human thymocyte globulin (MELD-ATG) is efficacy in Type 1 diabetes subjects. I have no more comments and suggestions for the study protocol. Because it is already very clear and detailed. The group and sample size are reasonable. The outcome is clear and right chosen. All the measurable lab test is sufficient and good to carry out. I just don't know if the journal would like to publish just a protocol like this to your readers?

Reviewer: 1

Competing interests of Reviewer: No

We thank the reviewer for their kind comments on our study protocol.

Reviewer: 2

Dr. Richard Leslie, Queen Mary University of London

Comments to the Author:

bmjopen-2021-053669

Immunotherapy has been used with some success in type 1 diabetes, One immunosuppressive agent that has shown promise in subjects aged 12-45 years is anti-thymocyte globulin (ATG), Thymoglobuline®, and further exploration in lower age groups and at lower doses is warranted.

The study MELD-ATG is a phase 2, multi-centre, randomised, double-blind, placebo- controlled, multi-arm parallel-group trial in participants (n=114) 5-25 years diagnosed with T1D within 3– 9 weeks of planned treatment day 1. Participants are being randomised to placebo, 2.5 mg/kg, 1.5 mg/kg, 0.5 mg/kg and 0.1 mg/kg ATG total dose in a 1:1:1:1:1 allocation ratio. The next six cohorts of 12-15 participants will be randomised to placebo, 2.5 mg/kg, and one or two selected middle ATG total doses in a 1:1:1:1 or 1:1:1 allocation ratio, as dependent on the number of middle doses, given intravenously over 2 consecutive days.

The primary objective will be to determine the changes in stimulated C-peptide response over the first 2 hours of a mixed meal tolerance test (MMTT) at 12 months for 2.5mg/kg ATG arm versus the placebo. Conditional on finding a significant difference at 2.5 mg/kg, a minimally effective dose will be sought. Secondary objectives include the determination of the effects of a particular ATG treatment dose on 1) stimulated C-peptide, 2) HbA1c, 3) daily insulin dose, 4) time in range by intermittent continuous glucose monitoring (CGM) measures, 5) fasting and stimulated dry blood spot (DBS) C-peptide measurements.

1. Strengths The study protocol and the participants are well versed in these trials and the format has been extensively used under the INNODIA umbrella. The project objective is worthwhile, the aim to examine if younger subjects and lower doses can be used successfully are fully justified in the opinion of this reviewer. Inclusion and exclusion criteria are well established plus a position on live vaccines .

We thank the reviewer for their kind comments on our study protocol.

2. Weakness It may not work. That aside the authors are conversant with the many issues such studies raise and have provided a protocol that is extensively tested in this area.

We note the reviewer's comment and thank them for their faith in our protocol in this new study.

3. Tragically David Dunger recently died so they will need to alter the email.
We have done so on page 1 of the resubmitted Main document (page 2 of the marked copy). Charlotte Wilhelm-Benartzi will now be the corresponding author, having drafted the original paper.
4. Novel features in this study include the use of time in range and dried blood spot c-peptide and glucose. This exploration study is worthwhile but at a preliminary stage in development.
We note the reviewer's comment and agree that these measures are indeed preliminary; however as the reviewer stated their exploration is worthwhile, hence our inclusion of these as secondary objectives.
5. The reviewer notes with approval that exploratory mechanistic studies are to be performed.
We thank the reviewer for their comment on our exploratory mechanistic studies.

Reviewer: 2

Competing interests of Reviewer: none

Reviewer: 3

Dr. Yasmine Elhenawy, Ain Shams University

Comments to the Author:

MELD-ATG is a phase II, dose ranging, efficacy study of (ATG) in new onset T1D; including participants 5-25 years. The study is the first evaluating efficacy and safety of ATG among patients aged 5-11 years and the first as well trying to identify the minimally effective low dose of ATG (MELD).

An important well designed interventional study that will provide possible data on the safety, tolerability, and efficacy of 2.5 mg/kg ATG and even lower doses in pediatric, adolescent and adult population with newly diagnosed T1D. Early Intervention with preservation of beta- cells of the pancreas represent a paradigm shift in managing T1D.

Reviewer: 3

Competing interests of Reviewer: nothing to declare

We thank the reviewer for their kind comments on our study protocol.